# Predicting antimicrobial resistance in *Pseudomonas aeruginosa* with machine learning-enabled molecular diagnostics

Ariane Khaledi[1,2,†], Aaron Weimann[2,3,4,†] (iD), Monika Schniederjans[1,2,‡], Ehsaneddin Asgari[3,5,‡],
Tzu-Hao Kuo[3], Antonio Oliver[6], Gabriel Cabot[6], Axel Kola[7], Petra Gastmeier[7], Michael Hogardt[8],
Daniel Jonas[9], Mohammad RK Mofrad[5,10], Andreas Bremges[3,4] (iD), Alice C McHardy[3,4,§,*] (iD) &
Susanne Häussler[1,2,§,**] (iD)

## Abstract

Limited therapy options due to antibiotic resistance underscore the need for optimization of current diagnostics. In some bacterial species, antimicrobial resistance can be unambiguously predicted based on their genome sequence. In this study, we sequenced the genomes and transcriptomes of 414 drug-resistant clinical *Pseudomonas aeruginosa* isolates. By training machine learning classifiers on information about the presence or absence of genes, their sequence variation, and expression profiles, we generated predictive models and identified biomarkers of resistance to four commonly administered antimicrobial drugs. Using these data types alone or in combination resulted in high (0.8–0.9) or very high (> 0.9) sensitivity and predictive values. For all drugs except for ciprofloxacin, gene expression information improved diagnostic performance. Our results pave the way for the development of a molecular resistance profiling tool that reliably predicts antimicrobial susceptibility based on genomic and transcriptomic markers. The implementation of a molecular susceptibility test system in routine microbiology diagnostics holds promise to provide earlier and more detailed information on antibiotic resistance profiles of bacterial pathogens and thus could change how physicians treat bacterial infections.

**Keywords** antibiotic resistance; biomarkers; clinical isolates; machine learning; molecular diagnostics

**Subject Categories** Biomarkers; Chromatin, Transcription & Genomics; Microbiology, Virology & Host Pathogen Interaction

## Introduction

The rise of antibiotic resistance is a public health issue of greatest importance (Cassini *et al*, 2019). Growing resistance hampers the use of conventional antibiotics and leads to increased rates of ineffective empiric antimicrobial therapy. If not adequately treated, infections cause suffering, incapacity, and death, and impose an enormous financial burden on healthcare systems and on society in general (Alanis, 2005; Gootz, 2010; Fair & Tor, 2014). Despite growing medical need, FDA approvals of new antibacterial agents have substantially decreased over the last 20 years (Kinch *et al*, 2014). Alarmingly, there are only few agents in clinical development for the treatment of infections caused by multidrug-resistant Gram-negative pathogens (Bush & Page, 2017).

*Pseudomonas aeruginosa*, the causative agent of severe acute as well as chronic persistent infections, is particularly problematic. The opportunistic pathogen exhibits high intrinsic antibiotic resistance and frequently acquires resistance-conferring genes via

1  Department of Molecular Bacteriology, Helmholtz Centre for Infection Research, Braunschweig, Germany
2  Molecular Bacteriology Group, TWINCORE-Centre for Experimental and Clinical Infection Research, Hannover, Germany
3  Computational Biology of Infection Research, Helmholtz Centre for Infection Research, Braunschweig, Germany
4  German Center for Infection Research (DZIF), Braunschweig, Germany
5  Molecular Cell Biomechanics Laboratory, Departments of Bioengineering and Mechanical Engineering, University of California, Berkeley, CA, USA
6  Servicio de Microbiología y Unidad de Investigación Hospital Universitario Son Espases, Instituto de Investigación Sanitaria Illes Balears (IdISPa), Palma de Mallorca, Spain
7  Institute of Hygiene and Environmental Medicine, Charité – Universitätsmedizin Berlin, Berlin, Germany
8  Institute of Medical Microbiology and Infection Control, University Hospital Frankfurt, Frankfurt/Main, Germany
9  Faculty of Medicine, Institute for Infection Prevention and Hospital Epidemiology, Medical Center-University of Freiburg, Freiburg, Germany
10  Molecular Biophysics and Integrated Bioimaging Division, Lawrence Berkeley National Lab, Berkeley, CA, USA
  *Corresponding author. Tel: +49 531 391 55271; E-mail: Alice.McHardy@helmholtz-hzi.de
  **Corresponding author. Tel: +49 531 6181 3000; E-mail: Susanne.Haeussler@helmholtz-hzi.de
  †These authors contributed equally to this work as the first authors
  ‡These authors contributed equally to this work as the second authors
  §Shared last authors

horizontal gene transfer (Lister *et al*, 2009; Partridge *et al*, 2018). Furthermore, the accelerating development of drug resistance due to the acquisition of drug resistance-associated mutations poses a serious threat.

The lack of new antibiotic options underscores the need for optimization of current diagnostics. Diagnostic tests are a core component in modern healthcare practice. Especially in light of rising multidrug resistance, high-quality diagnostics becomes increasingly important. However, to provide information as the basis for infectious disease management is a difficult task. Antimicrobial susceptibility testing (AST) has experienced little change over the years. It still relies on culture-dependent methods, and as a consequence, clinical microbiology diagnostics is labor-intensive and slow. Culture-based AST requires 48 h (or longer) for definitive results, which leaves physicians with uncertainty about the best drugs to prescribe to individual patients. This delay also contributes to the spread of drug resistance (Oliver *et al*, 2015; López-Causapé *et al*, 2018).

The introduction of molecular diagnostics could become an alternative to culture-based methods and could be critical in paving the way to fight antimicrobial resistance. Identification of genetic elements of antimicrobial resistance promises a deeper understanding of the epidemiology and mechanisms of resistance and could lead to a timelier reporting of the resistance profiles as compared to conventional culture-based testing. It has been demonstrated that for a number of bacterial species, antimicrobial resistance can be highly accurately predicted based on information derived from the genome sequence (Gordon *et al*, 2014; Bradley *et al*, 2015; Moradigaravand *et al*, 2018). However, in the opportunistic pathogen *P. aeruginosa* even full genomic sequence information is insufficient to predict antimicrobial resistance in all clinical isolates (Kos *et al*, 2015). *Pseudomonas aeruginosa* exhibits a profound phenotypic plasticity mediated by environment-driven flexible changes in the transcriptional profile (Dötsch *et al*, 2015). For example, *P. aeruginosa* adapts to the presence of antibiotics with the overexpression of the *mex* genes, encoding the antibiotic extrusion machineries MexAB-OprM, MexCD-OprJ, MexEF-OprN, and MexXY-OprM. Similarly, high expression of the *ampC*-encoded intrinsic beta-lactamase confers antimicrobial resistance (Haenni *et al*, 2017; Juan *et al*, 2017; Goli *et al*, 2018; Martin *et al*, 2018). Those transcriptional responses are frequently fixed in clinical *P. aeruginosa* strains, e.g., due to mutations in negative regulators of gene expression (Frimodt-Møller *et al*, 2018; Juarez *et al*, 2018). Thus, the isolates develop an environment-independent resistance phenotype. Up-regulation of intrinsic beta-lactamases as well as overexpression of efflux pumps that contribute to the resistance phenotype makes gene-based testing a challenge, because it is difficult to predict from the genomic sequence, which (combinations of) mutations would lead to an up-regulation of resistance-conferring genes (Llanes *et al*, 2004; Fernández & Hancock, 2012; Schniederjans *et al*, 2017).

In this study, we investigated whether we can reliably predict antimicrobial resistance in *P. aeruginosa* using not only genomic but also quantitative gene expression information. For this purpose, we sequenced the genomes of 414 drug-resistant clinical *P. aeruginosa* isolates and recorded their transcriptional profiles. We built predictive models of antimicrobial susceptibility/resistance to four commonly administered antibiotics by training machine learning classifiers. From these classifiers, we inferred candidate marker panels for a diagnostic assay by selecting resistance- and susceptibility-informative markers via feature selection. We found that the combined use of information on the presence/absence of genes, their sequence variation, and gene expression profiles can predict resistance and susceptibility in clinical *P. aeruginosa* isolates with high or very high sensitivity and predictive value.

# Results

### Taxonomy and antimicrobial resistance distribution of 414 DNA- and mRNA-sequenced clinical *Pseudomonas aeruginosa* isolates

A total of 414 *P. aeruginosa* isolates were collected from clinical microbiology laboratories of hospitals across Germany and at sites in Spain, Hungary, and Romania (Fig 1A). For all isolates, the genomic DNA was sequenced and transcriptional profiles were recorded. This enabled us to use not only the full genomic information but also information on the gene expression profiles as an input to machine learning approaches.

We inferred a maximum likelihood phylogenetic tree based on variant nucleotide sites (Fig 1B). The tree was constructed by mapping the sequencing reads of each isolate to the genome of the *P. aeruginosa* PA14 reference strain and then aligning the consensus sequences for each gene. The isolates exhibited a broad taxonomic distribution and separated into two major phylogenetic groups. One included PAO1, PACS2, LESB58, and a cluster of high-risk clone ST175 isolates; the other included PA14, as well as one large cluster of high-risk clone ST235 isolates. Both groups comprised several further clades with closely related isolates of the same sequence type as determined by multilocus sequencing typing (MLST).

Next, we recorded antibiotic resistance profiles for all isolates regarding the four common anti-pseudomonas antimicrobials, tobramycin (TOB), ceftazidime (CAZ), ciprofloxacin (CIP), and meropenem (MEM) (Bassetti *et al*, 2018; Cardozo *et al*, 2019; Tümmler, 2019) using agar dilution method. Most isolates of our clinical isolate collection exhibit antibiotic resistance against these four antibiotics (Fig 1C, Dataset EV1). One-third had a multidrug-resistant (MDR) phenotype, defined as non-susceptible to at least three different classes of antibiotics (Magiorakos *et al*, 2012).

### Machine learning for predicting antimicrobial resistance

We used the genomic and transcriptomic data of the clinical *P. aeruginosa* isolates to infer resistance and susceptibility phenotypes to ceftazidime, meropenem, ciprofloxacin, and tobramycin with machine learning classifiers. For each antibiotic, we included all respective isolates categorized as either "resistant" or "susceptible". For the genomic data, we included sequence variations (single nucleotide polymorphisms; SNPs, including small indels) and gene presence or absence (GPA) as features. In total, we analyzed 255,868 SNPs, represented by 65,817 groups with identical distributions of SNPs across isolates for the same group, and 76,493 gene families with presence or absence information, corresponding to 14,700 groups of identically distributed gene families. 1,306 of these gene families had an indel in some isolate genomes, which we included as an additional feature. We evaluated SNP and GPA

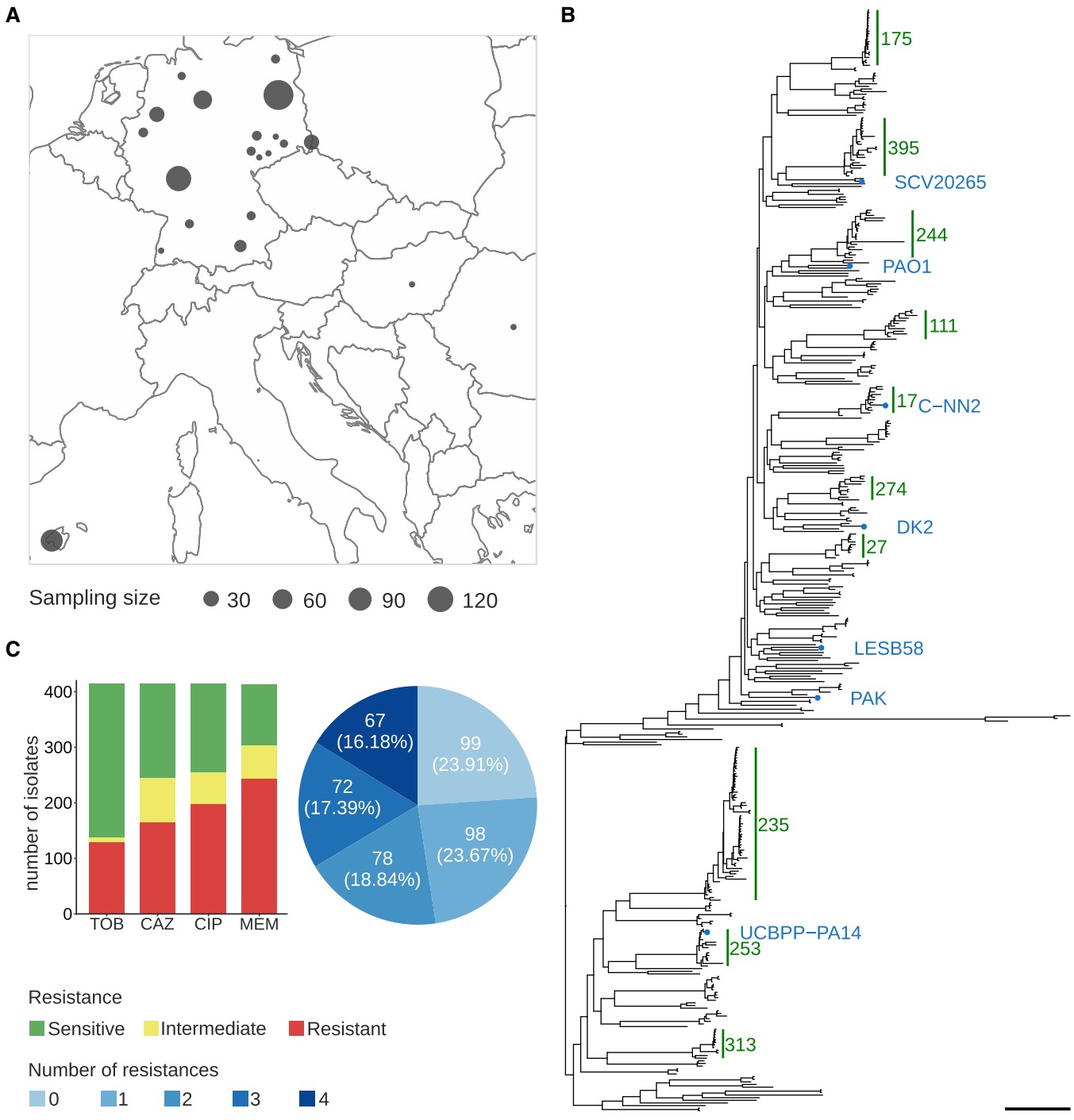

**Figure 1. Geographic and phylogenetic distribution of 414 clinical *Pseudomonas aeruginosa* isolates used in this study.**

A  Geographic sampling site distribution, where circle size is proportional to the number of isolates from a particular location.

B  Phylogenetic tree of the clinical isolates and seven reference strains (blue dots). A PA7-like outgroup clade including two clinical isolates is not shown. Abundant high-risk clones are indicated by green bars. Scale bar: 0.04.

C  Antimicrobial susceptibility profiles against the four commonly administered antibiotics tobramycin (TOB), ceftazidime (CAZ), ciprofloxacin (CIP), and meropenem (MEM) determined by agar dilution according to Clinical & Laboratory Standards Institute Guidelines (CLSI, 2018).

groups in combination with gene expression information for 6,026 genes (Fig 2).

For each drug, we randomly assigned isolates to a training set that comprised 80% of the resistant and susceptible isolates, respectively,

and the remaining 20% to a test set. Parameters of machine learning models were optimized on the training set and their value assessed in cross-validation, while the test set was used to obtain another independent performance estimate. As bacterial population structure

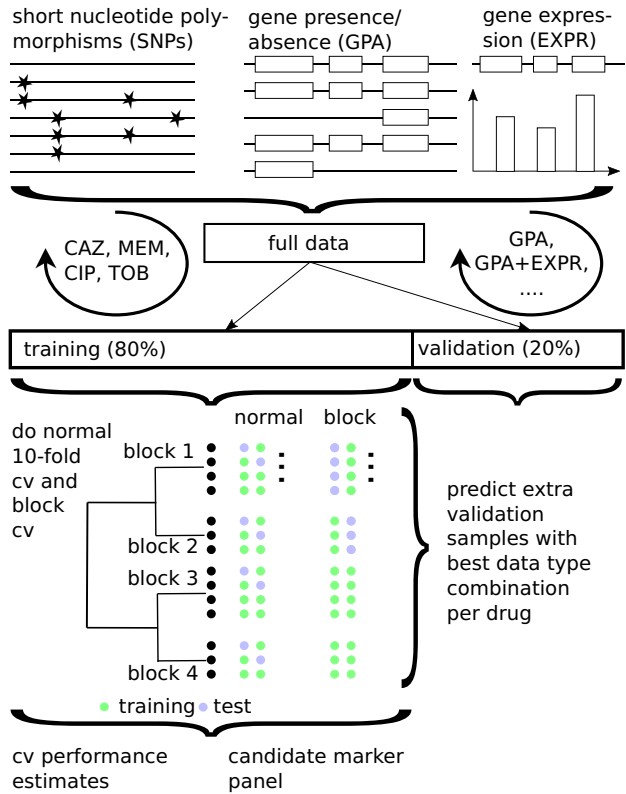

short nucleotide poly-morphisms (SNPs)

gene presence/absence (GPA)

gene expres-sion (EXPR)

CAZ, MEM, CIP, TOB

full data

GPA, GPA+EXPR, ....

training (80%)

validation (20%)

do normal 10-fold cv and block cv

normal  block

block 1

block 2
block 3

block 4

predict extra validation samples with best data type combination per drug

● training ● test

cv performance estimates

candidate marker panel

**Figure 2.  Training and validating a diagnostic classifier for antimicrobial susceptibility prediction for four different drugs based on genomic (GPA/SNPs) and transcriptomic profiles (EXPR).**

The best data type combination was determined using 80% of the data in standard and phylogenetically informed cross-validation (cv) and further validated on the remaining 20% of the data.

can influence machine learning outcomes, e.g., it has been shown before in *Escherichia coli* that phylo-groups' specific markers alone could be used to predict antibiotic resistance phenotypes with accuracies of 0.65–0.91, depending on the antibiotic (Moradigaravand *et al*, 2018), we also assessed performance while accounting for population structure based on sequence types through a block cross-validation approach. We trained several machine learning classification methods on SNPs, GPA, and expression features individually and in combination for predicting antibiotic susceptibility or resistance of isolates and evaluated the classifier performances. We determined MIC (minimal inhibitory concentration) values of all clinical isolates with agar dilution according to CLSI guidelines (CLSI, 2018) to use as the gold standard for evaluation purposes.

We calculated the sensitivity and predictive value of resistance (R) and susceptibility (S) assignment, as well as the macro F1-score, as an overall performance measure based on a classifier trained on a specific data type combination. The sensitivity reflects how good that classifier is in recovering the assignments of the underlying gold standard, representing the fraction of susceptible, or resistant, samples, respectively. The predictive value reflects how trustworthy the assignments of this particular classifier are, representing the fraction of correct assignments of all susceptible or resistant

assignments, respectively. The F1-score is the harmonic mean of the sensitivity and predictive value for a particular class, i.e., susceptible or resistant. The macro F1-score is the average over the two F1-scores.

We used the support vector machine (SVM) classifier with a linear kernel, as in Weimann *et al* (2016), to predict sensitivity or resistance to four different antibiotics. Parameters were optimized in nested cross-validation, and performance estimates averaged over five repeats of this setup. The combined use of (i) GPA, (ii) SNPs, and (iii) information on gene expression resulted in high (0.8–0.9) or very high (> 0.9) sensitivity and predictive values (Fig 3). Notably, the relative contribution of the different information sources to the susceptibility and resistance sensitivity strongly depended on the antibiotic. To assess the effect of the classification technique, we compared the performance of an SVM classifier with a linear kernel to that of random forests and logistic regression, which we and others have successfully used for related phenotype prediction problems (Asgari *et al*, 2018; Her & Wu, 2018; Wheeler *et al*, 2018). For this purpose, we used the data type combination with the best macro F1-score in resistance prediction with the SVM. We evaluated the classification performance in nested cross-validation and on a held-out test dataset. In addition, we performed a phylogeny-aware partitioning of our dataset, to assess the phylogenetic generalization ability of our technique.

The performance of the SVM in random cross-validation was comparable to logistic regression (macro F1-score for the SVM: $0.83 \pm 0.06$ vs. logistic regression: $0.84 \pm 0.06$), but considerably better than the random forest classifiers ($0.67 \pm 0.14$; Appendix Figs S1 and S2, Dataset EV2). The performance on the held-out dataset was in a comparable range (SVM: $0.87 \pm 0.07$; logistic regression: $0.90 \pm 0.04$; random forest $0.71 \pm 0.16$). We furthermore observed similar macro F1-scores inferred in the phylogenetically selected cross-validation (SVM: $0.87 \pm 0.07$; logistic regression: $0.86 \pm 0.07$; random forest $0.72 \pm 0.13$), which suggests only a minor influence of the bacterial phylogeny on the classification performance. The performance on the phylogenetically selected held-out dataset was again comparable, though performance for the random forest deteriorated in comparison with the cross-validation results (SVM: $0.86 \pm 0.06$; logistic regression $0.83 \pm 0.06$; random forests $0.56 \pm 0.03$).

Ciprofloxacin resistance and susceptibility based on SVMs could be correctly predicted with a sensitivity of $0.92 \pm 0.01$ and $0.87 \pm 0.01$, and with simultaneously high predictive values of $0.91 \pm 0.01$ and $0.90 \pm 0.01$, respectively, using solely SNP information. The sensitivity of $0.80 \pm 0.04$ and $0.79 \pm 0.02$ and predictive value of $0.73 \pm 0.01$ and $0.76 \pm 0.02$ to predict ciprofloxacin susceptibility and resistance based exclusively on gene expression data were also high. However, there was no added value of using information on gene expression in addition to SNP information for the prediction of susceptibility/resistance toward ciprofloxacin.

For the prediction of tobramycin susceptibility and resistance, the machine learning classifiers performed almost equally well when the three input data types (SNPs, GPA, and gene expression) were used individually (values > 0.8). SNP information was predictive of tobramycin resistance; however, it did not further improve the classification performance when combined with the other data types. GPA information alone was the most important data type for classifying tobramycin resistance and susceptibility providing

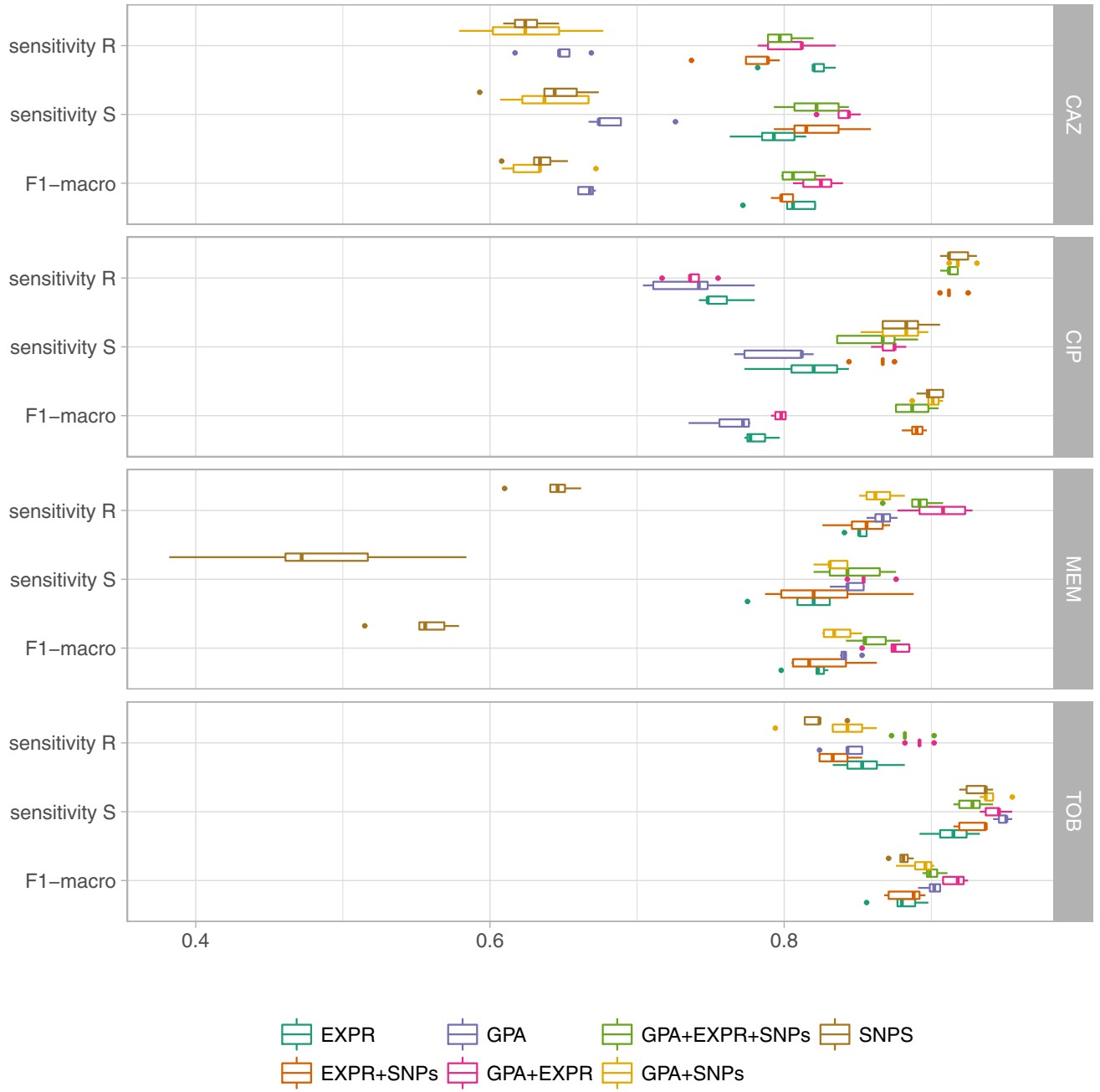

**Figure 3. Evaluation of AMR classification with a support vector machine (R: resistant; S: susceptible) using different performance metrics and data types (EXPR: gene expression; GPA: gene presence or absence; and SNPs: single nucleotide polymorphisms) or combinations thereof.**

Each individual panel depicts the results for one of four different anti-pseudomonal antibiotics (CAZ, CIP, MEM, and TOB). The solid vertical line in the box plots represents the median, the box limits depict the 25th and 75th percentile, and the lower and upper hinges include values within ± 1.5 times the interquartile range. Values outside that range were plotted as solid dots.

sensitivity values of $0.84 \pm 0.01$ and $0.95 \pm 0.01$ and predictive values of $0.88 \pm 0.01$ and $0.93 \pm 0.01$, respectively. The performance of GPA-based prediction increased further when gene expression values were included ($P$-value of a one-sided $t$-test: $-0.0069$ based on the macro F1-score as determined in repeated cross-validation; sensitivity values of $0.89 \pm 0.01$ and $0.94 \pm 0.01$ for resistance and susceptibility prediction, respectively, and predictive values of $0.88 \pm 0.01$ and $0.95 \pm 0.01$).

For the correct prediction of meropenem resistance/susceptibility, gene presence/absence was most influential (sensitivity values of $0.87 \pm 0.01$ and $0.84 \pm 0.01$ for resistance and susceptibility prediction, respectively, and predictive values of $0.92 \pm 0.00$ and $0.74 \pm 0.01$). As observed for tobramycin, the use of genome-wide information on GPA and of information on gene expression in combination increased the sensitivity to detect resistance as well as susceptibility to meropenem to $0.91 \pm 0.02$ and $0.86 \pm 0.01$ and

Table 1. Performance of support vector machine (SVM) classifier to predict sensitivity or resistance to four different antibiotics.

| Antibiotic | Markers used | Sensitivity (resistance) | Sensitivity (susceptibility) | Predictive value (resistance) | Predictive value (susceptibility) | F1-score | Number of markers* |
|---|---|---|---|---|---|---|---|
| CAZ | GPA+EXPR | 0.83 ± 0.02 | 0.81 ± 0.02 | 0.81 ± 0.02 | 0.83 ± 0.01 | 0.82 ± 0.01 | 37 |
| TOB | GPA+EXPR | 0.89 ± 0.01 | 0.94 ± 0.01 | 0.88 ± 0.01 | 0.95 ± 0.01 | 0.92 ± 0.01 | 59 |
| MEM | GPA+EXPR | 0.91 ± 0.02 | 0.86 ± 0.01 | 0.93 ± 0.01 | 0.81 ± 0.03 | 0.87 ± 0.01 | 93 |
| CIP | SNPs | 0.92 ± 0.01 | 0.87 ± 0.01 | 0.91 ± 0.01 | 0.90 ± 0.01 | 0.90 ± 0.01 | 50 |

*The number of markers indicates the number of (combined) features that resulted in the least complex SVM model within one standard deviation of the peak performance, i.e., with the best macro F1-score and as few as possible features for each drug.

the predictive values to 0.93 ± 0.01 and 0.81 ± 0.03, respectively (*P*-value of a one-sided *t*-test: 0.004).

For ceftazidime, using only information on gene presence/absence revealed a sensitivity of susceptibility/resistance prediction of 0.69 ± 0.01 and 0.66 ± 0.01, and predictive values of 0.66 ± 0.01 and 0.67 ± 0.01, respectively. Adding gene expression information considerably improved the performance of susceptibility and resistance sensitivity to 0.83 ± 0.02 and 0.81 ± 0.02 and predictive values of 0.81 ± 0.02 and 0.83 ± 0.01 (*P*-value of a one-sided *t*-test $7.1 \times 10^{-7}$). In summary, for tobramycin, ceftazidime, and meropenem combining GPA and expression information gave the most reliable classification results, whereas for ciprofloxacin we found that only using SNPs provided the best performance (Table 1 and Dataset EV3). Thus, for the remainder of the manuscript, we will focus on the results obtained with classifiers trained on those data type combinations.

### A candidate drug resistance marker panel

We determined the minimal number of molecular features required to obtain the highest macro F1-score for each drug. We inferred the number of features contributing to the classification from the number of non-zero components of the SVM weight vectors, using a standard cross-validation setup. For each value of the C parameter, which controls the amount of regularization imposed on the model, the cross-validation procedure was repeated five times (Fig 4, Dataset EV4). Performance of antimicrobial resistance prediction peaked for the candidate classifiers using between 50 and 100 features. Notably, the ciprofloxacin classifier required only two SNPs until the learning curve performance was almost saturated, whereas classifiers of drugs that included expression and gene presence/absence markers required more features (> 50) to reach saturation.

Next, we determined the C parameter resulting in the least complex SVM model within one standard deviation of the peak performance, i.e., with the best macro F1-score and as few as possible features for each drug (Friedman *et al*, 2001). We chose our candidate marker panel for each drug as the set of all non-zero features and designated the respective model as the most suitable diagnostic classifier. We used SNP information for ciprofloxacin resistance and susceptibility prediction and the combination of GPA and expression features for tobramycin, meropenem, and ceftazidime. We refer to each of these classifiers as the candidate classifier for susceptibility and resistance prediction for a particular drug.

The ciprofloxacin candidate marker panel contained 50 SNPs. The meropenem, ceftazidime, and tobramycin marker lists consisted of 93, 37, and 59 expression and GPA features. The complete list of candidate markers for the prediction of resistance against the four antibiotics is given in Dataset EV5. This list includes the candidate markers of the three input features namely GPA, gene expression, and SNPs alone and in combination. Table 2 is a shortlist of the panel markers for each drug based on the data combination that had allowed us to train the most reliable classifier.

To test the performance of the candidate marker panel-based classifiers on an independent set of clinical *P. aeruginosa* isolates, we used them to predict antibiotic resistance for the samples of the test dataset (Fig 5, Dataset EV6). On this held-out data, we obtained an F1-sore for all drugs that was similarly high as before: Namely this was 0.95 for meropenem, 0.77 for ceftazidime, and 0.96 for tobramycin, using gene expression and gene presence/absence features, and 0.87 for ciprofloxacin using SNP information. These results indicate that the diagnostic classifiers have good generalization abilities when applied to new samples. We observed more variability across drugs than in nested cross-validation, which is expected due to the smaller size of the test set.

### Improvement of assignment accuracy with increasing sample numbers

We next investigated how prediction performance depended on the number of samples used for classifier training. We trained the SVM classifiers on random subsamples of different sizes of the full dataset with 414 isolates. For each model, we recorded the macro F1-score in five repeats of 10-fold nested cross-validation (Fig 6). The classification performance saturates for all our classifiers well before using all available training samples, suggesting that when adding more isolates for resistance classification, the classification performance would improve only very slowly. Markers potentially remaining undiscovered in our study might have very small effect sizes, requiring much larger dataset sizes for their detection. Interestingly, the number of samples required until the performance curve plateaued depends on the drugs and data types used. For ciprofloxacin, the performance of susceptibility/resistance prediction based on SNPs saturated quickly, likely due to the large impact of the known mutations in the quinolone resistance-determining region (QRDR), whereas the classifiers for the other three drugs, which were trained on expression and gene presence/absence information, required more samples until the F1-score plateaued. For these classifiers, the dispersion of the macro F1-score for subsets of the data with fewer samples is also considerably higher than for the ciprofloxacin SNP models.

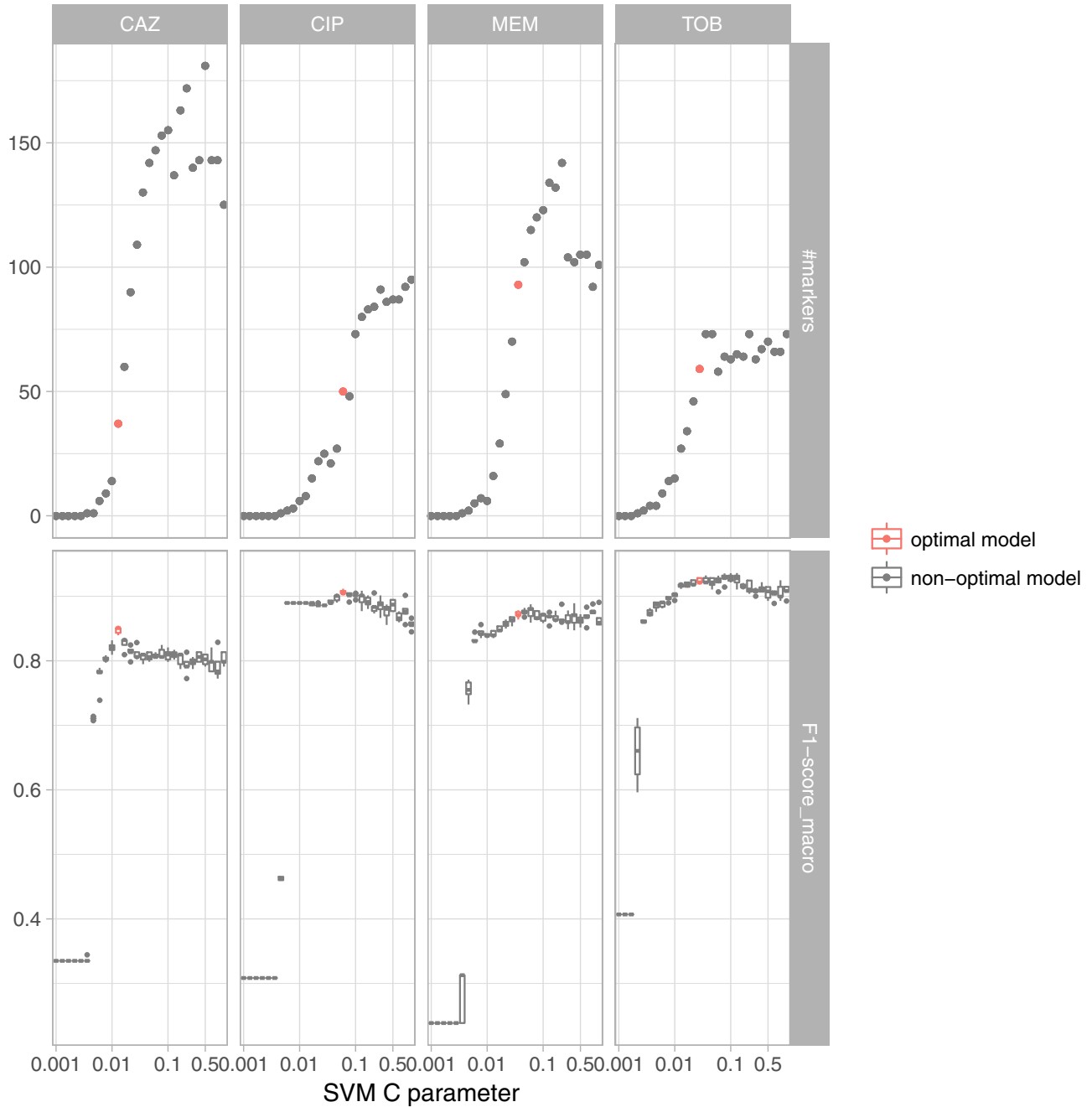

**Figure 4. The number of features used by the support vector machine classifier (top panels) and corresponding classification performance (bottom panels) varies with the hyperparameter C.**

The C parameter is inversely related to the number of markers being included in the model, i.e., lower values for the C parameter yield models with less features. The SVM resistance/susceptibility classifier was evaluated in five repeats of 10-fold nested cross-validation. Each panel depicts the results for a different drug (CAZ, CIP, MER, and TOB) based on the best data type combination (GPA+EXPR/SNPs). The model with the fewest features within one standard deviation of the maximal performance was selected as the most suitable diagnostic classification model (red) (Dataset EV5). The solid vertical line in the box plots represents the median, the box limits depict the 25th and 75th percentile, and the lower and upper hinges include values within ± 1.5 times the interquartile range. Values outside that range were plotted as solid dots.

**Performance estimation stratifying by sequence type suggests some influence of the bacterial phylogeny on the prediction**

In *P. aeruginosa*, different phylo-groups might contain different antibiotic resistance genes or mutations alone or in combinations. Thus, if there was an association of distinct resistance-conferring genes with certain phylo-groups, our machine learning approach might identify markers that distinguish between different phylo-groups rather than between susceptible and resistant clinical isolates. In Figs EV1–EV4, we show

**Table 2.** The top 15 candidate markers ranked according to the contribution of each marker to the support vector machine classifier for each drug based on the best performing combination of data types.

| Drug | Data type | PA14/CARD gene_id | PA14/CARD gene_acc | Prokka/Roary gene_id | SNP position |
|---|---|---|---|---|---|
| TOB (GPA_EXPR) | GPA | A7J11_00271 | qacEdelta1 | emrE | |
| | GPA | A7J11_02078 | sul1 | folP_2_indel | |
| | GPA | PA14_04410 | ptsP | ptsP | |
| | GPA | | | group_282 | |
| | GPA | PA14_20840 | | group_14073 | |
| | GPA | | | group_20477 | |
| | EXPR | PA14_15450 | traJ | | |
| | GPA | PA14_15100 | | mepM_1 | |
| | GPA | A7J11_02078 | sul1 | folP_2 | |
| | GPA | | | group_8948 | |
| | GPA | | | group_51714 | |
| | EXPR | PA14_38410 | amrB | | |
| | GPA | PA14_18565 | alg8 | alg8 | |
| | GPA | | | group_3462 | |
| | GPA | | | group_17749 | |
| MEM (GPA_EXPR) | GPA | | | group_596_indel/oprD_1 | |
| | GPA | PA14_51880 | oprD | oprD_4_indel | |
| | EXPR | PA14_46070 | gbuA | | |
| | EXPR | PA14_05550 | oprM | | |
| | GPA | | | group_3638 | |
| | EXPR | PA14_51880 | oprD | | |
| | GPA | | | group_6217/pknK_1 | |
| | EXPR | PA14_05540 | mexB | | |
| | EXPR | PA14_07630 | | | |
| | EXPR | PA14_63090 | lldD | | |
| | GPA | PA14_11960 | | yabI_indel | |
| | EXPR | PA14_70940 | betA | | |
| | GPA | | | group_6280 | |
| | GPA | | | group_15876 | |
| | GPA | | | group_10960 | |
| CIP (SNPs) | SNP | PA14_23260 | gyrA | | 2015001 |
| | SNP | PA14_65605 | parC | | 5845617 |
| | SNP | PA14_55600 | | | 4947631 |
| | SNP | PA14_56040 | | | 5004892 |
| | SNP | PA14_30960 | traG | | 2690138 |
| | SNP | PA14_31010 | | | 2694327 |
| | SNP | PA14_29390 | | | 2545634 |
| | SNP | PA14_41560 | nasA | | 3710561 |
| | SNP | PA14_18260 | fruK | | 1567193 |
| | SNP | PA14_30910 | trbE | | 2685860 |
| | SNP | PA14_30960 | traG | | 2689741 |
| | SNP | PA14_59210 | | | 5274257 |
| | SNP | PA14_44640 | | | 3974007 |
| | SNP | PA14_41110 | | | 3665768 |
| | SNP | PA14_15460 | merA | | 1310089 |

**Table 2** (continued)

| Drug | Data type | PA14/CARD gene_id | PA14/CARD gene_acc | Prokka/Roary gene_id | SNP position |
|---|---|---|---|---|---|
| CAZ (GPA_EXPR) | EXPR | PA14_10790 | ampC | | |
| | GPA | A7J11_02078 | sul1 | folP_2 | |
| | GPA | | | group_8955 | |
| | EXPR | PA14_48900 | | | |
| | GPA | PA14_00810 | | group_13626 | |
| | EXPR | PA14_15770 | | | |
| | GPA | | | group_3462 | |
| | GPA | PA14_33690 | pudE | yojI | |
| | GPA | | | group_23010 | |
| | GPA | | | petE_indel | |
| | EXPR | PA14_31240 | | | |
| | EXPR | PA14_53500 | | | |
| | GPA | | | group_5517 | |
| | GPA | PA14_22650 | | group_14516_indel | |
| | GPA | | | group_63043 | |

For gene presence/absence (GPA) markers, we provide the gene id and accession based on the PA14 reference genome gene family member or based on the Comprehensive Antibiotic Resistance Database (CARD) (Jia et al, 2017). Otherwise, we include the gene name or id of each marker as generated by the bacterial genome annotation tool Prokka (Seemann, 2014) and protein family clustering software Roary (Page et al, 2015). Expression markers are based on the PA14 genome, too. For short nucleotide polymorphisms (SNPs), we report the genome position in the reference PA14 genome.

susceptibility and resistance of each isolate in the context of the phylogenetic tree as predicted by the diagnostic classifier and based on AST for each of the drug. To assess whether our predictive markers are biased by the phylogenetic structure of the clinical isolate collection, we assessed classification robustness in a block cross-validation approach. Here, isolates of phylo-groups with differing sequence types as determined by MLST were grouped into blocks and all isolates of a given block were only allowed to be either in the training or test folds (Figs 2 and 5). In addition, instead of using a random assignment of strains into test and training dataset, we analyzed the performance only allowing strains in a test dataset corresponding to the block cross-validation training dataset with sequence types that were not already included in this training dataset. For all classifiers including our candidate diagnostic classifiers, we found that the block cross-validation performance estimates were slightly lower than those obtained using a sequence type-unaware estimation (F1-score difference between ∼ 0.03 and 0.05 for the diagnostic classifiers). This was particularly apparent for some suboptimal data type combinations, such as for predicting tobramycin resistance using SNPs or gene expression, where a substantially lower discriminative performance was achieved in block- compared to random cross-validation (macro F1-score difference > 0.1, Dataset EV3). Interestingly, we observed that the ranking of the performance by data type remained almost identical for all drugs. Overall, the performance estimates we obtained using this phylogenetically insulated test dataset were comparable to the block cross-validation estimates, only tobramycin resistance prediction using classifiers trained fully or partly on SNPs dropped considerably in performance.

In summary, this confirmed that the various P. aeruginosa phylogenetic subgroups possess similar mechanisms and molecular markers for the resistance phenotype and that the identified markers are largely distinctive for resistance/susceptibility instead of phylogenetic relationships using most data type combinations. Despite the observed independence of the presence of genetic resistance markers and bacterial phylogeny, for some antibiotics and data types we also found a non-negligible phylo-group-dependent performance effect. This underlines the importance of assessing the impact of the phylogeny on the antimicrobial resistance prediction.

## Misclassified isolates are more frequent near the MIC breakpoints

We tested whether we could detect an overrepresentation of misclassified samples among the samples with a MIC value close to the breakpoints compared to samples with higher or lower MIC values, selecting samples from equidistant intervals (in log space) around the breakpoint. We report only the strongest overrepresentation for each drug after multiple testing correction. For ciprofloxacin, significantly more samples with a MIC between 0.5 and 8 were misclassified (31 of 139 samples (22%)) than samples with a MIC smaller than 0.5 or larger than 8 (7 of 219 samples (3%)) (Fisher's exact test with an FDR-adjusted $P$-value of $6.2 \times 10^{-8}$; Fig 7). For ceftazidime, we found that 46 of 177 samples (26%) with a MIC between 4 and 64 were misclassified whereas only 21 of 157 (13%) of samples with a MIC smaller or higher than those values were misclassified (adjusted $P$-value: 0.014). For meropenem, we found that 26 of 207 samples (13%) with a MIC between 1 and 16 were misclassified, but only 8 of 147 (5%) of all samples with a MIC smaller or higher than those values were misclassified (adjusted $P$-value: 0.05). For tobramycin, no significant difference was found.

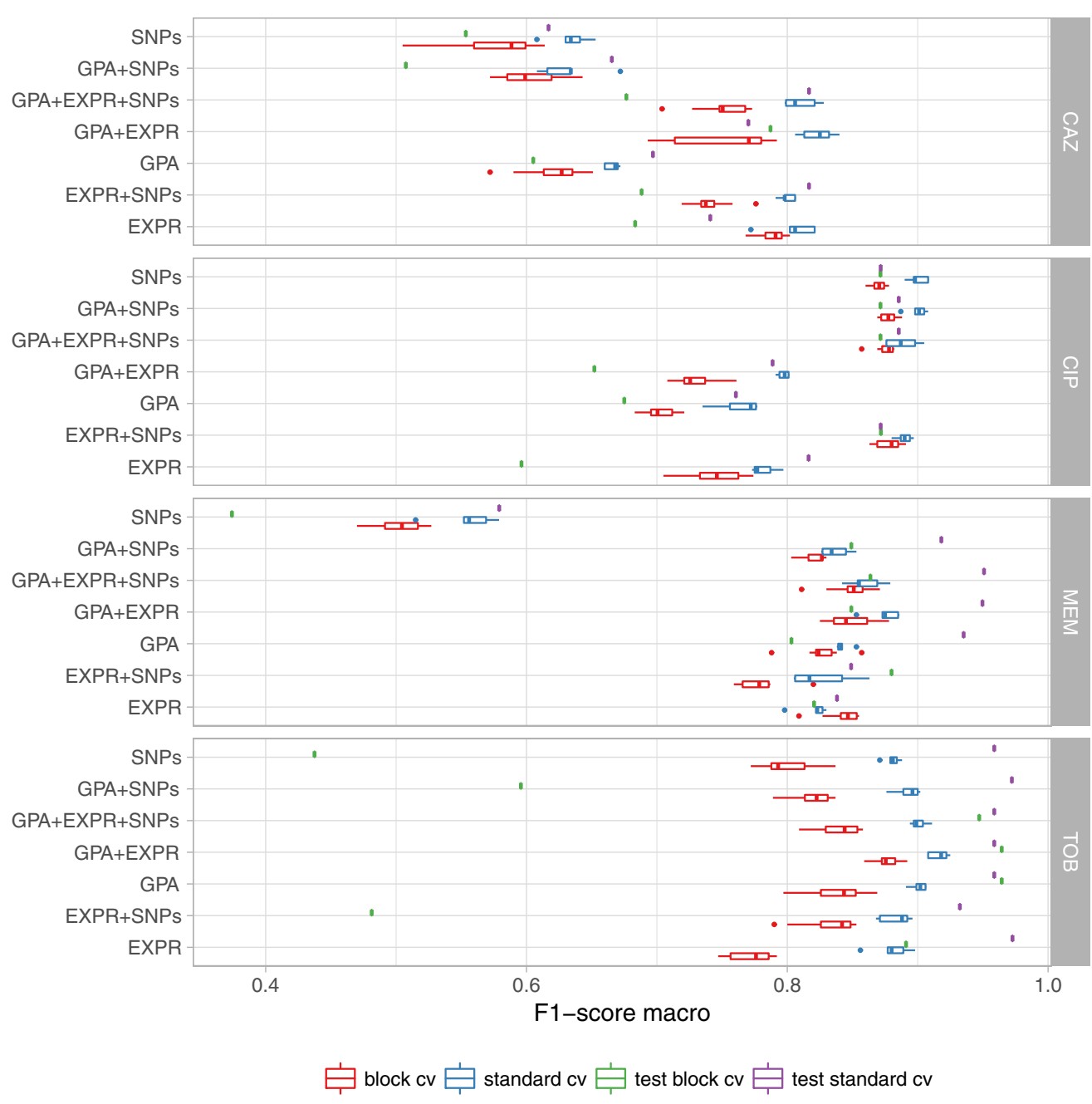

**Figure 5. Performance of the support vector machine (SVM) classifier for antimicrobial resistance and susceptibility prediction for different data types, different drugs, and different evaluation schemes.**

The SVM performance was summarized by the F1-score and is shown for standard cross-validation (standard_cv, blue) and cross-validation using phylogenetically related blocks of isolates (block_cv, red) based on the training dataset (80% of the isolates) and for the validation dataset (green; 20% of the isolates). EXPR: gene expression; GPA: gene presence and absence with indel information. SNPs: short nucleotide polymorphisms. The solid vertical line in the box plots represents the median, the box limits depict the 25th and 75th percentile, and the lower and upper hinges include values within ± 1.5 times the interquartile range. Values outside that range were plotted as solid dots.

## Discussion

One of the most powerful weapons in the battlefield of drug-resistant infections is rapid diagnostics of resistance. Earlier and more detailed information on the pathogens' antimicrobial resistance profile has the potential to change antimicrobial prescribing behavior and improve the patient's outcome. The demand for faster results has initiated investigation of molecular alternatives to today's culture-based clinical microbiology procedures. However, for the successful implementation of robust and reliable molecular tools, it is critical to identify

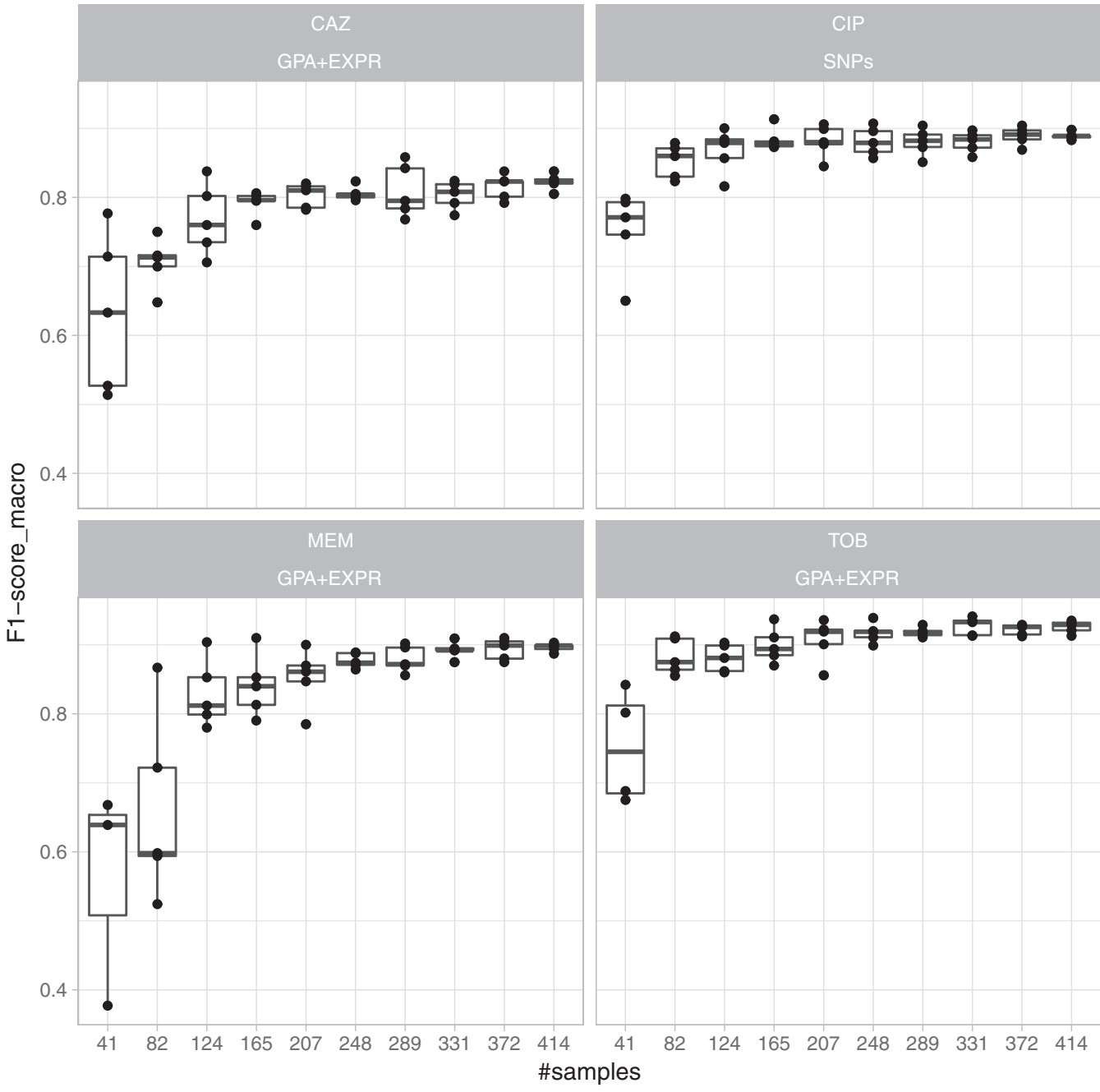

**Figure 6. Classification performance improves and plateaus with the number of training samples used.**

A support vector machine-based resistance/susceptibility classifier was trained on differently sized and randomly drawn subsamples from our isolate collection and evaluated in five repeats of a 10-fold nested cross-validation. Each panel depicts the results for a different drug (CAZ, CIP, MEM, and TOB) based on the best data type combination (GPA+EXPR/SNPs). The solid vertical line in the box plots represents the median, the box limits depict the 25th and 75th percentile, and the lower and upper hinges include values within $\pm$ 1.5 times the interquartile range. Values outside that range were plotted as solid dots.

the entirety of the molecular determinants of resistance. Failure to detect resistance can lead to the administration of ineffective or suboptimal antimicrobial treatment. This has direct consequences for the patient and poses significant risks especially in the critically ill patient. Conversely, failing to identify susceptibility may result in the avoidance of a drug despite the fact that it would be suitable to treat the pathogen, in the extreme case leading to patient death due to a

lack of known treatment options. Overtreatment could also be a consequence and the needless use of broad-spectrum antibiotics. This drives costs in the hospital, puts patients at risk for more severe side effects, and may contribute to the development of drug resistance by applying undesired selective pressures.

In this study, we show that without any prior knowledge on the molecular mechanisms of resistance, machine learning approaches

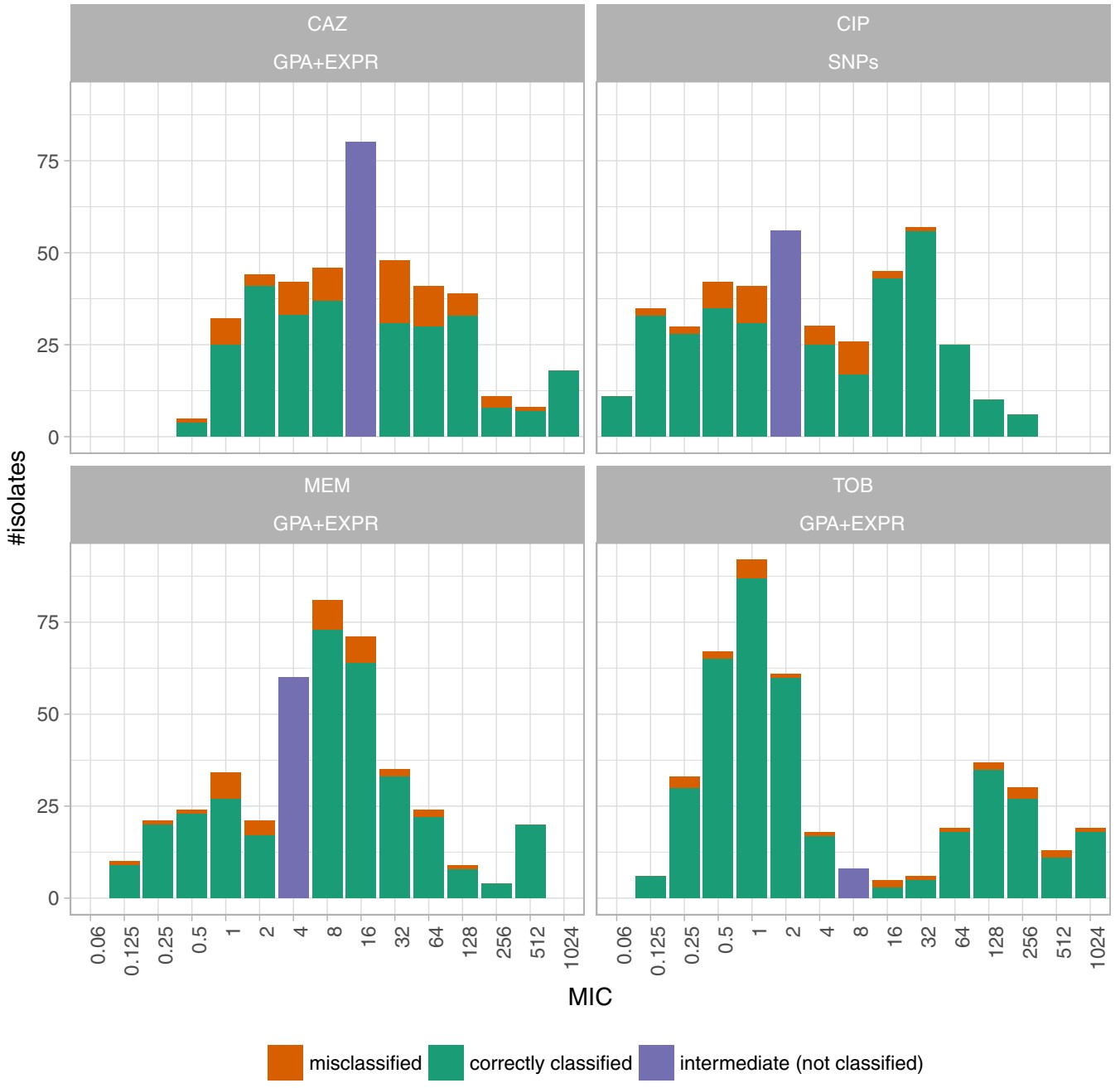

**Figure 7. Number of samples misclassified and correctly predicted by the support vector machine resistance and susceptibility classifier (SVM) grouped by their minimum inhibitory concentration.**

Each panel depicts the results for a different anti-pseudomonal drug (CAZ: ceftazidime; CIP: ciprofloxacin; MEM: meropenem; TOB: tobramycin) for the best data type combination (GPA+EXPR/SNPs). Misclassified and correctly classified samples for the training dataset (80%) were inferred in a 10-fold cross-validation. An SVM trained on the training dataset was used to predict resistance/susceptibility of the test samples (20%). The number of misclassified samples in the training (80%) and test set was aggregated.

using genomic and transcriptomic features can provide high antibiotic resistance assignment capabilities for the opportunistic pathogen *P. aeruginosa*. The performance of drug resistance prediction was strongly dependent on the antibiotic.

Ciprofloxacin resistance and susceptibility prediction mostly relied on SNP information. Particularly, two SNPs in the quinolone resistance-determining region (QRDR) of *gyrA* and *parC* had the

strongest impact on the classification (Dataset EV3). This is an expected finding as quinolone antibiotics act by binding to their targets, gyrase, and topoisomerase IV (Bruchmann *et al*, 2013); and target-mediated resistance caused by specific mutations in the encoding genes is the most common and clinically significant form of resistance (del Barrio-Tofiño *et al*, 2017). Although the sensitivity to predict resistance and susceptibility from only gene expression data

were also high toward ciprofloxacin, there was no added value of using information on gene expression in addition to SNP information. Nevertheless, for the design of a diagnostic test system, it might be of value to include also gene expression information as a fail-safe strategy. Interestingly, among the gene expression classifiers that were associated with ciprofloxacin susceptibility/resistance, we found *prtN,* which is involved in pyocin production. Enhanced pyocin production is, as the SOS response, induced under DNA-damaging stress conditions (Migliorini *et al*, 2019) and was recently reported to contribute to ciprofloxacin resistance (Fan *et al*, 2019).

For the prediction of tobramycin susceptibility and resistance, the machine learning classifiers performed almost equally well when the three input data types (SNPs, GPA, and gene expression) were used individually (sensitivity and predictive values > 0.8). Remarkably, the combined use of the GPA and the gene expression datasets improved the classification performance. Although SNP information also was predictive of tobramycin resistance, it did not further improve the classification performance when combined with the other feature types. GPA information alone was the most important data type for classifying tobramycin resistance or susceptibility. The majority of aminoglycoside-resistant clinical isolates harbor genes encoding for aminoglycoside-modifying enzymes (AMEs). The AMEs are very diverse but are usually encoded by genes located on mobile genetic elements, including integrons and transposons. In accordance, the presence of respective markers that indicate the presence of these mobile elements was found to be strongly associated with tobramycin resistance (e.g., *qacEdelta1, sul1,* or *folP*). However, the most influential discriminator was the presence of the *emrE* gene. *EmrE* has been described to directly impact on aminoglycoside resistance by mediating the extrusion of small polyaromatic cations (Li *et al*, 2003). Second, we identified the presence of *ptsP* (encoding phosphoenolpyruvate protein phosphotransferase) as an important marker for tobramycin resistance. This gene has previously already been associated with tobramycin resistance in a transposon mutant library screen (Schurek *et al*, 2008).

The performance of GPA-based prediction increased further when gene expression values were included. We found, e.g., *amrB* (*mexY*), which encodes a multidrug efflux pump known to confer to aminoglycoside resistance (Westbrock-Wadman *et al*, 1999; Lau *et al*, 2014), as one of the top candidates within the marker panel. This confirms that expression of efflux pumps is an important bacterial trait that drives the resistance phenotype in *P. aeruginosa.* Tobramycin resistance/susceptibility was also associated with an altered expression or SNPs within genes involved in type 4 pili motility (*pilB pilV2, pilC,* and *pilH*) and the type three secretion system (*pcr* genes). Although the connection to tobramycin resistance might be not exactly obvious, it has been proposed that surface motility can lead to extensive multidrug adaptive resistance as a result of the collective dysregulation of diverse genes (Sun *et al*, 2018).

For the correct prediction of meropenem resistance/susceptibility, gene presence/absence was most influential. Interestingly, in contrast to tobramycin resistance classification, we observed a substantial accumulation of indels in specific marker genes. Among these marker genes were *ftsY,* involved in targeting and insertion of nascent membrane proteins into the cytoplasmic membrane, *czcD,* encoding a cobalt–zinc–cadmium efflux protein, and *oprD.*

Inactivation of the porin OprD is the leading cause of carbapenem non-susceptibility in clinical isolates (Köhler *et al*, 1999). As expected, also a decreased *oprD* gene expression in the resistant group of isolates was identified as an important discriminator. Interestingly though, the most important gene expression marker was not the down-regulated *oprD*, but an up-regulation of the gene *gbuA,* encoding a guanidinobutyrase in the arginine dehydrogenase pathway, in the meropenem-resistant group of isolates. It is known that arginine metabolism plays a critical role during host adaptation and persistence (Hogardt & Heesemann, 2013). Interestingly, it was also described before that GbuA is linked to virulence factor expression and the production of pyocyanin (Jagmann *et al*, 2016). Our results indicate that up-regulation of *gbuA* might be the result of a non-fully functional OprD porin. Since OprD has been shown to be involved in arginine uptake (Tamber & Hancock, 2006), one might speculate that lack of arginine due to a non-functional OprD triggers the expression of *gbuA* to compensate for the fitness defect of the *oprD* mutant.

Furthermore, components encoding the MexAB-OprM efflux pump (*mexB, oprM*) were identified as important features associated with resistance. This efflux pump is known to export beta-lactams, including meropenem (Li *et al*, 1995; Srikumar *et al*, 1998; Clermont *et al*, 2001).

As observed for tobramycin, the correct prediction of ceftazidime resistance/susceptibility was strongly influenced by both gene expression values (here *ampC, fpvA, pvdD,* and *algF*) and gene presence/absence (including the presence of mobile genetic elements). While AmpC is a known intrinsic beta-lactamase, able to hydrolyze cephalosporins (Lister *et al*, 2009), the association of ceftazidime resistance with expression variations in *fpvA, pvdD,* and *algF,* involved in the uptake of iron and the production of alginate, respectively, is less clear. Interestingly, sequence variations in regulators such as AmpR, AmpG, AmpD (including AmpD homologs), and *mpl* and alteration in penicillin-binding proteins such as PBP4 (*dacB*) have been described to trigger constitutive *ampC* overexpression (Bagge *et al*, 2002; Juan *et al*, 2005, 2006; Schmidtke & Hanson, 2008; Moya *et al*, 2009; Balasubramanian *et al*, 2012; Cabot *et al*, 2018). AmpR, however, does not only control *ampC* expression but has also been described to be a global regulator of resistance and virulence in *P. aeruginosa* and to be an important acute–chronic switch regulator (Balasubramanian *et al*, 2015). As such, AmpR is also involved in the regulation of alginate production as well as iron acquisition via siderophores. This might explain why expression of *fpvA, pvdD,* and *algF* was found to be associated with ceftazidime resistance.

Since we did not identify any of the previously described sequence variations in the various regulators of *ampC* expression by the use of the machine learning approach, we re-analyzed them in more detail. Interestingly, we identified a small number of isolates in the resistant group (11 of 165) harboring an R504C substitution in the gene ftsI (PBP3). Mutations in PBP3 have been described to represent an AmpC-independent resistance evolution *in vitro* and occur upon beta-lactam treatment *in vivo* (Cabot *et al*, 2016, 2018; López-Causapé *et al*, 2017). Particularly, the R504C substitution has been found in clinical cystic fibrosis isolates and is contributing to ceftazidime resistance (López-Causapé *et al*, 2017). However, all but three of our CAZ-resistant isolates with a R504C mutation in *ftsI* likewise showed a strong *ampC* overexpression, most likely

explaining why *ftsI* was not identified as a discriminative marker in our analysis, despite clearly harboring resistance-associated mutations.

Adding information on the gene expression considerably improved the performance of susceptibility and resistance sensitivity for ceftazidime, which was not observed in a similar scale for any other antibiotic.

Interestingly, although we recognized widely overlapping resistance profiles for all antibiotics (Fig EV5), we did not observe a strong co-resistance bias in the identified markers. For example, among the best performing classifiers for meropenem, ceftazidime, and tobramycin, there were only overlapping markers between ceftazidime and tobramycin. These included expression of PA14_15420 and presence of A7J11_02078/sul1/folP_2, group_282, group_3462, and group_5517 which account for 5/59 and 5/37 of the total features or 14.7%/17.1% of the total weight of the ceftazidime and tobramycin SVM classifiers, respectively. Group_282, group_3462, and group_5517 genes are hypothetical genes. Sul1, which is located on mobile elements (usually class 1 integrons), could indicate that the shared signal of the tobramycin and ceftazidime classifiers is due to resistance genes being found on the same resistance cassettes, as class 1 integrons carrying beta-lactamases as well as aminoglycoside-modifying enzymes are frequently detected (Poirel *et al*, 2001; Fonseca *et al*, 2005).

In conclusion, we demonstrate that extending the genetic features (SNPs and gene presence/absence) with gene expression values is key to improving performance. Thereby, relative contribution of the different categories of biomarkers to the susceptibility and resistance sensitivity strongly depended on the antibiotic. This is in stark contrast to the prediction of antibiotic resistance in many Enterobacteriaceae, where knowledge of the presence of resistance-conferring genes, such as beta-lactamases, is usually sufficient to correctly predict the susceptibility profiles. However, analysis of the gene expression marker lists revealed that the resistance phenotype in the opportunistic pathogen *P. aeruginosa* (and possibly also in other non-fermenters) is multifactorial and that alterations in gene expression can alter the resistance phenotype quite substantially.

Intriguingly, we found that the performance of our classifiers improved if the isolates exhibited MIC values that were not close to the breakpoint. This was especially apparent for ciprofloxacin. It has been demonstrated that patients treated with levofloxacin for bloodstream infections caused by Gram-negative organisms for which MICs were elevated, yet still in the susceptible category, had worse outcomes than similar patients infected with organisms for which MICs were lower (Defife *et al*, 2009). A possible explanation for treatment failure could be the presence of first-step mutations in *gyrA* that lead to MIC values near the breakpoint. If subjected to quinolones, those isolates can rapidly acquire second-step mutations in *parC* that would then exhibit a fully resistant phenotype. An additional explanation might also be that generally, MICs have a low level of reproducibility (Turnidge & Paterson, 2007; Juan *et al*, 2012; Javed *et al*, 2018). A non-accurate categorization due to uncertainty in testing near the MIC breakpoint can explain failure in the assignment of drug resistance by the machine learning classifiers.

Capturing the full repertoire of markers that are relevant for predicting antimicrobial resistance in *P. aeruginosa* will require further studies, to expand the predictive power of the established marker lists. The remaining misclassified samples in our study on

the basis of these marker lists represent a valuable resource to uncover further spurious resistance mutations.

The broad use of molecular diagnostic tests promises more detailed and timelier information on antimicrobial-resistant phenotypes. This would enable the implementation of early and more targeted, and thus more effective antimicrobial therapy for improved patient care. Importantly, a molecular assay system can easily be expanded to test for additional information such as the clonal identity of the bacterial pathogen or the presence of critical virulence traits. Thus, availability of molecular diagnostic test systems can also provide prognostic markers for disease outcome and give valuable information on the clonal spread of pathogens in the hospital setting. However, to realize the full potential of the envisaged molecular diagnostics, clinical studies will be needed to demonstrate that broad application of such test systems will have an impact in clinical decision-making, provide the basis for more efficient antibiotic use, and also decrease the costs of care.

## Materials and Methods

### Strain collection and antibiotic resistance profiling

Our study included 414 clinical *P. aeruginosa* isolates provided by different clinics or research institutions: 350 isolates were collected in Germany (138 at the Charité Berlin (CH), 89 at the University Hospital in Frankfurt (F), 39 at the Hannover Medical School (MHH), and 84 at different other locations). Sixty-two isolates were provided by a Spanish strain collection located at the Son Espases University Hospital in Palma de Mallorca (ESP), and two samples originated from Hungary and Romania, respectively.

All clinical isolates were tested for their susceptibility toward the four common anti-pseudomonas antibiotics tobramycin (TOB), ciprofloxacin (CIP), meropenem (MEM), and ceftazidime (CAZ). Minimal inhibitory concentration (MIC) testing and breakpoint determination were performed in agar dilution according to Clinical & Laboratory Standards Institute (CLSI) Guidelines (CLSI, 2018). MIC testing was performed in triplicates for all isolates. If results varied, up to five replicates were used. Only isolates with at least three matching results were included in the study. Most of the isolates were categorized as multidrug-resistant (resistant to three or more antimicrobial classes, Dataset EV1). As reference for differential gene expression and sequence variation analysis, the UCBPP-PA14 strain was chosen.

### Colony screening

To rule out possible contaminations, all isolates were continuously re-streaked at least twice from single colonies. Only isolates with reproducible outcomes in phenotypic tests were included in the final panel, which furthermore passed DNA sequencing quality control (> 85% sequencing reads mapped to *P. aeruginosa* UCBPP-PA14 reference genome, total read GC content of 64–66%).

### RNA sequencing

For comparable whole-transcriptome sequencing, all clinical isolates and the UCBPP-PA14 reference strain were cultivated at 37°C in LB

broth and harvested in RNAprotect (Qiagen) at $OD_{600} = 2$. Sequencing libraries were prepared using the ScriptSeq RNA-Seq Library Preparation Kit (Illumina), and short read data (single end, 50 bp) were generated on an Illumina HiSeq 2500 machine creating on average 3 million reads per sample. The 414 samples were distributed across 24 independent sequencing pools. We assessed possible batch effects using triplicates of the PA14-wt (Appendix Fig S3). The majority of the genome was very stably expressed across the replicates (Pearson correlation coefficient $\geq 0.94$).

The reads were mapped with Stampy [v1.0.23; (Lunter & Goodson, 2011)] to the UCBPP-PA14 reference genome (NC_008463.1), which is available for download from the Pseudomonas Genome database (http://www.pseudomonas.com). Mapping and calculation of reads per gene (rpg) values were performed as described previously (Khaledi *et al*, 2016). Expression counts were log-transformed (to deal with zero values, we added one to the expression counts).

### DNA sequencing

Sequencing libraries were prepared from genomic DNA using the NEBNext Ultra DNA Library Prep Kit (New England Biolabs) and sequenced in paired-end mode on Illumina HiSeq or MiSeq machines, generating either $2 \times 250$ or $2 \times 300$ bp reads. On average, 2.89 million reads were generated per isolate (ranging from 653,062 to 21,086,866 reads with at least 30 times total genome coverage per isolate). All reads were adapter and quality-clipped using fastq-mcf (Andrews, 2010).

### SNP calling

DNA sequencing reads were mapped with *Stampy* as described above (see RNA sequencing). For variant calling, SAMtools, v0.1.19 (Li *et al*, 2009), was used. We noticed that sometimes sequencing errors (particularly around indels) tended to influence calling accuracy (e.g., a SNP was called although the nucleotide chance appeared only in a fraction of the reads). For correction of these obvious errors, we implemented an additional step where nucleotide positions were converted into the most likely sequence according to the most frequently occurring nucleotide at this position.

### Phylogeny

Paired-end reads (read length 150, fragment size 200) of eight reference genomes were simulated using art_illumina (v2.5.8) with the default error profile at 20-fold coverage (Huang *et al*, 2012). Together with our 414 clinical isolates, the sequencing reads were mapped to the coding regions of reference genome UCBPP-PA14 by BWA-MEM (v0.7.15) (preprint: Li, 2013). SAMtools (v1.3.1) (Li *et al*, 2009) and BamTools (Barnett *et al*, 2011) (v2.3.0) were used for indexing and sorting the aligned reads, respectively, followed by variant calling using FreeBayes (v1.1.0) (preprint: Garrison & Marth, 2012). The consensus coding sequences were computed by BCFtools (v1.6) (Li, 2011) and then sorted into families by corresponding reference regions. A gene family was excluded if the gene sequence of any of its member differed by more than 10% in lengths as compared to the length of the reference genome gene family. Totally, 5,936 families were

retained. The sequences of each family were aligned by MAFFT (v7.310) (Katoh & Standley, 2013), and the alignments were concatenated. SNP sites that were only present in a single isolate were removed from the alignment. The final alignment was composed of 558,483 columns, and the approximately maximum likelihood phylogeny was then inferred by FastTree (v2.1.10, double precision) (Price *et al*, 2010).

### Pan-genome analysis and indel calling

The trimmed reads were assembled with SPAdes, v.3.0.1, using the –careful parameter (Bankevich *et al*, 2012). The assembled genomes were annotated using Prokka (v1.12) (Seemann, 2014) using the metagenome mode of Prokka for gene calling, as we had noticed that genes on resistance cassettes were often missed by the standard isolate genome gene calling procedure. The gene sequences were clustered into gene families using Roary (Page *et al*, 2015). We observed that Roary frequently clustered together gene sequences of drastically varying lengths due to indels or start and stop codon mutations in those gene sequences and frequently also splits orthologous genes into more than one gene family. To overcome this behavior, we modified Roary to require at least 95% alignment coverage in the BLAST step (https://github.com/hzi-bifo/Roary).

For matching the Prokka annotation and the reference annotation of the *PA14* strain, we used bedtools (Quinlan, 2014) to search for exact overlaps of the gene coordinates. In a second step, we identified all Roary gene families that contained a *PA14* gene. To identify insertions and deletions in the Roary gene families, we extracted nucleotide sequences for each gene family and used MAFFT (Katoh & Standley, 2013) to infer multiple sequence alignments. We restricted this analysis to gene families present in at least 50 strains. Then, we used MSA2VCF (https://github.com/lindenb/jvarkit/) for calling variants in the gene sequences and restricted the output to insertion and deletions of at least nine nucleotides.

### Support vector machine classification

For applying cross-validation, the dataset was split once randomly and once phylogenetically informed (see below) into $k$-folds ($k$ set to 10, unless specified otherwise). Classifier hyperparameters were optimized on a $k - 1$ fold-sized partition, and performance of the optimally parameterized method was determined on the left out $k$ fraction of the data. This was performed for all possible $k$ partitions, assignments summarized, and final performance measures obtained by averaging.

### Comparison of different machine learning classifiers

We used the training set for hyperparameter tuning of the classifiers, i.e., a linear SVM, RF, and LR, optimizing the F1-score in 10-fold cross-validation and then evaluated the best trained classifier on the held-out set. The expression features (EXPR) and any combination of features with another data type (GPA and SNPs) were transformed to have zero mean and unit variance, whereas binary features (GPA, SNPs, and GPA+SNPs) were not transformed. The RF classifier was optimized for the macro F1-score over different

**The paper explained**

**Problem**

Limited therapy options due to the emergence and spread of multi-drug resistance leave clinicians with uncertainty about which drug to prescribe. Inadequate initial therapy, however, may cause suffering or death of infected patients, promotes further resistant development, and imposes an enormous financial burden on healthcare systems and on society in general.

**Results**

We integrated genomic, transcriptomic, and phenotypic data on antibiotic resistance profiles of 414 clinical *Pseudomonas aeruginosa* isolates and used a machine learning-based approach to identify sets of molecular markers that allowed a reliable prediction of antibiotic resistance against four antibiotic classes. Using information on (i) the presence or absence of genes, (ii) sequence variations within genes, and (iii) gene expression profiles alone or in combinations resulted in high (0.8–0.9) or very high (> 0.9) sensitivity and predictive values. Importantly, transcriptome data significantly improved the prediction outcome as compared to using genome information alone. Identified biomarkers included known antibiotic resistance determinants (e.g., *gyrA*, *ampC*, *oprD*, efflux pumps) as well as markers previously not associated with antibiotic resistance.

**Impact**

Our findings demonstrate that the identification of molecular markers for the prediction of antibiotic resistance holds promise to change current resistance diagnostics. However, gene expression information may be required for highly sensitive and specific resistance prediction in the problematic opportunistic pathogen *P. aeruginosa*.

hyperparameters: (i) the number of decision trees in the ensemble, (ii) the number of features for computing the best node split, (iii) the function to measure the quality of a split, and (iv) the minimum number of samples required to split a node. The logistic regression and the linear SVM were optimized for the macro F1-score over: (i) the C parameter (inverse to the regularization strength) and (ii) class weights (to be balanced based on class frequencies or to be uniform over all classes). Subsequently, we measured the performance of the optimized classifiers over accordingly generated, held-out sets of samples.

In clinical practice, *P. aeruginosa* strains isolated from patients are likely to include sequence types that are already part of our isolate collection. To obtain a more conservative estimate of the performance of the antimicrobial susceptibility prediction, we also validated the classifiers on a held-out dataset composed of entirely new sequence types and also selected the folds in cross-validation to be non-overlapping in terms of their sequence types (block cross-validation). For partitioning the isolate collection into sequence types, we used spectral clustering over the phylogenetic similarity matrix (preprint: von Luxburg, 2007). We obtained this matrix by applying a Gaussian kernel over the matrix of distances between isolates based on the branch lengths in the phylogenetic tree.

### Multilocus sequence typing

Consensus fastq files for each isolate were created with SAMtools to extract the seven *P. aeruginosa* relevant MLST gene sequences (*acsA, aroE, guaA, mutL, nuoD, ppsA,* and *trpE*). Sequence type

information was obtained from the *P. aeruginosa* MLST database (https://pubmlst.org/paeruginosa/; Jolley & Maiden, 2010).

### Implementation

We encapsulated the sequencing data processing routines in a stand-alone package named seq2geno2pheno. The SVM classification was conducted with Model-T, which is built on scikit-learn (Pedregosa *et al*, 2011) and was already used as the prediction engine in our previous work on bacterial trait prediction (Weimann *et al*, 2016). seq2geno2pheno also implements a framework to use a more broader set of classifiers, which we used to compare different classification algorithms for drug resistance prediction. Finally, we created a repository that includes scripts to re-produce the figures and analyses presented in this paper using the aforementioned packages.

## Data availability

- RNA-Seq data: Gene Expression Omnibus GSE123544 (http://www.ncbi.nlm.nih.gov/geo/query/acc.cgi?acc = GSE123544)
- Figure generation and analyses scripts: GitHub (https://github.com/hzi-bifo/Predicting_PA_AMR_paper)
- Sequencing data processing and classifier comparison software: GitHub (https://github.com/hzi-bifo/seq2geno2pheno)
- SVM classification software: GitHub (https://github.com/hzi-bifo/Model-T)
- DNA-Seq data: Sequence Read Archive PRJNA526797 (https://www.ncbi.nlm.nih.gov/sra/?term = PRJNA526797)
- Direct input for training the machine learning classifiers (genomic features and resistance data tables): Zenodo https://doi.org/10.5281/zenodo.3464542 (https://zenodo.org/record/3464542#.Xf YShRtCeUk)

Expanded View for this article is available online.

### Acknowledgements

Financial support from the European Research Council (http://erc.europa.eu/) (ERC COMBAT grant 724290) is gratefully acknowledged. This study was further supported by the German Federal Ministry of Education and Research (grant 01 KI 9907) and the German Centre for Infection Research (DZIF). Members of the study group on "Spread of nosocomial infections and resistant pathogens" contributed to bacterial isolates. GC and AO are supported by Instituto de Salud Carlos III, Ministerio de Economía, Industria y Competitividad, Spanish Network for Research in Infectious Diseases (REIPI RD16/0016), and grant PI18/00076. We thank Adrian Kordes for his assistance in preparing the DNA sequencing libraries and Agnes Nielsen for support in conducting the AST testing of the clinical isolates. We also thank Jürgen Tomasch for analyzing the correlation of the transcriptional profiles of the PA14-wt replicates.

### Author contributions

AKo, MH, PG, DJ, and GC generated data. AKh and MS performed experiments. AW, EA, MRKM, and ACM developed the computational methodology. AW, AKh, MS, EA, T-HK, AB, and MRKM analyzed the data. AW, AKh, MS, AO, ACM, and SH interpreted the results. SH and ACM conceived the project, designed

experiments, and supervised the work. AW, AKh, T-HK, and EA generated figures and tables. AKh, AW, SH, and ACM wrote the paper. All authors read and approved the final manuscript.

## Conflict of interest

The authors declare that they have no conflict of interest.

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
