## [Review Process File · EMBO Molecular Medicine]

Predicting antimicrobial resistance in *Pseudomonas aeruginosa* with machine learning-enabled molecular diagnostics

Ariane Khaledi, Aaron Weimann, Monika Schniederjans, Ehsaneddin Asgari, Tzu-Hao Kuo, Antonio Oliver, Gabriel Cabot, Axel Kola, Petra Gastmeier, Michael Hogardt, Daniel Jonas, Mohammad R.K. Mofrad, Andreas Bremges, Alice C. McHardy, Susanne Häussler

Review timeline:

Submission date:	23 May 2019
Editorial Decision:	8 July 2019
Revision received:	7 November 2019
Editorial Decision:	10 December 2019
Revision received:	24 December 2019
Accepted:	9 January 2020

Editor: Céline Carret

Transaction Report:

1st Editorial Decision

8 July 2019

Thank you for the submission of your manuscript to EMBO Molecular Medicine. We have now heard back from the three referees whom we asked to evaluate your manuscript.

You will see from the set of comments below that both referees are supportive of publication pending the following revision: controls, replicates and more details and explanations are needed. A more thorough description of the machine learning protocol is required as indeed referee 1 would like to get access to input data for the machine learning model so that s/he could try to replicate some of the results if possible. Alternatively, the analyses should be redone using more standard approaches. Finally, some part of the work should be re-written for a non-specialist audience.

We would therefore welcome the submission of a revised version within three months for further consideration and would like to encourage you to address all the criticisms raised as suggested to improve conclusiveness and clarity. Please note that EMBO Molecular Medicine strongly supports a single round of revision and that, as acceptance or rejection of the manuscript will depend on another round of review, your responses should be as complete as possible.

I look forward to receiving your revised manuscript.

***** Reviewer's comments *****

Referee #1 (Comments on Novelty/Model System for Author):

I believe that the data presented is of sufficient quality, but requires proper controls (especially on expression data) to ensure that it is relevant for the task and for future reuse. Improving on that would greatly improve the manuscript.

Referee #1 (Remarks for Author):

With antimicrobial resistance increasingly becoming a recognized public health threat, it is of great importance to improve the diagnostic tools available in the clinic, so that treatment decisions can be made quickly and precisely. This in turn would solve the problem of misuse of antibiotics, which is thought to be one of factors leading to the emergence of antimicrobial resistance. Using genetic and molecular markers has recently emerged as a promising way to accomplish this task, which could eventually become cost-effective as the price of recording molecular markers drops. In this manuscript, Khaledi, Weimann and colleagues use a collection of more than 400 *Pseudomonas aeruginosa* strains to test if a machine learning approach could be applied to predict resistance to four antimicrobials using both genetic and expression data. The authors report very high predictive power, with the most relevant features recapitulating known resistance mechanisms and claiming how expression data might be needed to obtain an effective predictor for three of the four antibiotics tested with their approach.

I appreciated reading this manuscript and I agree with the points raised in the introduction and discussion about the need for effective and quick diagnostic tests. I have however several concerns regarding the generation of the molecular data used in the predictors (specifically the expression data), the strategy used for training and main assessment of predictive power (specifically the influence of population structure) and with some of the terminology used, which is at times not following the standards found in the field of machine learning. These concerns are detailed below; while I think that some of those concerns are somewhat serious, I also believe that the authors can address them with minimal need for additional experimental work. I also want to congratulate the authors for choosing to make their software pipeline reproducible and transparent. I haven't had the time to properly inspect the code, but it looks appropriate at a superficial look.

Main concerns

* Expression data

While it seems like the expression data has been previously reported in a peer-reviewed publication (doi: 10.1038/s41396-019-0412-1), I was surprised to see how each strain was measured using a single technical and biological replicate. Could the authors comment on that and provide some indication on the reproducibility of the data, for instance by repeating the expression measurements for a few samples? The methods section also seems to not mention the potential for batch effects, which can arise if more than one sequencing run has been used or if different operators have performed RNA extraction at different times. Can the authors report more details of their experimental setup and what's the extent of potential batch effects on their data? I am also curious to better understand which treatment of their input data has been fed to their machine learning models: from the methods section it looks like log-transformed counts were used, which might introduce a series of complications when used in machine learning models. In fact, absolute transcription levels might not be indicative of the relevance of a gene towards resistance, and might introduce bias in the way the weights are assigned during model learning, especially when mixed together with the genetic data, which is binary in nature. Can the authors clarify exactly what transformation of the expression data was used in their model and whether using something like fold-changes with respect to the reference improves the model?

* Model training and evaluation

Another major concern regards the way the machine learning models were trained and evaluated. Previous studies where machine learning approaches were used to predict antibiotics resistance have pointed out how population structure alone might sometimes be sufficient to deliver an accurate prediction (doi: 10.1101/403204), or how it "spills" in the genetic data (e.g. through gene presence/absence profiles, doi: 10.1371/journal.pcbi.1006258); furthermore, as samples are not independent from each other (especially if very closely related), great care has to be followed to

define the held-out (or test) set. The main results showed by the authors (i.e. Figure 3) uses a test set that seems to have been generated by taking a random sample of all the strains used; while later the authors report having tested for the influence of population structure on their prediction (pages 21-22), they fail to report with sufficient emphasis the results (i.e. no figure). However, Figure 5 clearly shows how population structure might be influencing the models; in all instances, the "block cv" predictive performance is significantly lower than the "standard cv". The authors should better address this point and perhaps consider using a test set that is phylogenetically "insulated" from the training set (as done in *E. coli* doi: 10.1371/journal.pcbi.1006258). While I greatly appreciated the data presented in Table 1, which clearly indicates how only a small subset of features is needed for an effective prediction, I would like to point out how the authors should better address the probable redundancy in their input data; in particular, gene presence/absence profiles and expression patterns are probably partly correlated, since an absent gene is likely to also not be expressed. Apart from that, previous works have tried to reduce the number of features when using bacterial genomics data (e.g. doi: 10.1371/journal.pcbi.1006258 and doi: 10.1186/1471-2164-13-170). Would the results change significantly if the input data was preconditioned to remove highly correlated features? Finally, I believe that the authors should correct the terminology used; specifically, they refer to a validation set (e.g. in Figure 5) when they probably meant test set (i.e. held-out set).

* Other concerns

The authors claim that expression data improves predictive power for three antibiotics out of four tested; when looking at Figure 3 I however get the impression that expression data significantly improves prediction only for Ceftazidime. The authors should therefore correct their statement, unless some information is missing.

Minor comments

* Page 6: "common anti-pseudomonas antimicrobials, tobramycin (TOB), ceftazidime (CAZ), ciprofloxacin (CIP), and meropenem (MEM)", can the authors provide a reference for this statement?

* Figure 1: it's difficult to read the colors on the phylogenetic tree, even on a screen

* Figure 3: using many different metrics to measure predictive power could perhaps be relegated to a supplementary figure (the metrics seem to be highly correlated anyway); changing this figure to be have a similar layout as Figure 5 (using only the F1-score) might help readability

* I couldn't find the genomic data on either SRA/ENA, even though the authors provided an accession number (PRJNA526797). Is it currently under an embargo?

* I couldn't find the input data used for training and prediction in the provided repositories: could the authors provide them in the next release, so that different approaches could be tested?

Referee #2 (Remarks for Author):

The authors have carried out a very comprehensive study, incorporating genomic and transcriptional data on a very large number of isolates of *P. aeruginosa* and using machine learning, in order to improve tools for prediction of antibiotic resistance in this species. The approaches used by the authors, in particular the integration of gene expression analysis with genome sequence analysis but also the use of machine learning in the context of *P. aeruginosa* antibiotic resistance, are novel and give important new insights. In addition the authors have been very rigorous in developing and validating the approaches that they used. Consequently they have developed a tool that can predict resistance with high accuracy. My only real reservation regarding the manuscript is that a number of the genes revealed by machine learning as being good predictors of resistance/sensitivity are non-intuitive (and one or two that might be expected do not appear). This may well reflect a lack of knowledge of the basis of antibiotic resistance in *P. aeruginosa* rather than a flaw in the study design, but the manuscript would be strengthened through more discussion of the genes uncovered in the analysis. The article is generally clearly written but some parts especially the machine learning sections (p.8-14) would be challenging for much of the target audience (molecular biologists, clinicians and medical doctors).

1. Title. In my opinion the title does not accurately reflect the content of the manuscript, they are not "fighting resistance". The authors have used machine learning to develop tools for understanding the basis of antibiotic resistance in *Pseudomonas aeruginosa*, extending to predicting resistance, and in my opinion a title along those lines would better reflect the manuscript contents. Using the tools developed here to fight resistance (or at least to provide better targeted treatment) is the next step!
2. Figure 1 shows that the authors have a very extensive diversity of isolates which is essential for the study. Some of the isolates were phylogenetically very close which runs the risk of mis-identifying resistance- or sensitivity- associated SNPs due to co-inheritance in clonal isolates. The authors' "block" approach showed that this was not a problem but it was a bit confusing that this is shown diagrammatically in Fig 2 but not explained until p. 22.
3. Along similar lines - Fig 1C shows the frequencies of resistance of isolates to different antibiotics. For isolates with multiple resistances was there a bias in co-resistances? Eg. for bacteria that were meropenem resistant, was there a disproportionate number that were also ceftazidime-resistant? A bias in co-resistance could mislead association between genome markers, or expression, and resistance to a given antibiotic.
4. Page 8 line 13-14. Having a portion of the isolates as a training set and a portion as the validation set is very logical, but how was the ratio of 80:20 decided on, did the authors investigate other ratios?
5. Figure 3 and Figure 5 - the multiple box plots and colours make this difficult to see in detail. The authors could try having black lines with coloured fill for the boxes.
6. Figure 4. This is one example where the non-specialist may struggle (this one did). It would be helpful to expand the explanation of the C parameter and the role of regularization. How do the data show that the ciprofloxacin classifier needed only 2 SNPs to saturate the learning curve whereas the other antibiotics needed 50 or more? - I can make a guess but it would be helpful to explain.
7. Table 2 and associated analyses. This contains a few surprises. For example, with tobramycin, a recent paper involving one of the authors indicated that *fusA1* was a major determinant of tobramycin resistance in *P. aeruginosa* (Lopez-Causape et al, AAC, 2018). It therefore seems surprising that *fusA1* SNPs did not strengthen the analysis. Similarly, ceftazidime resistance is associated with mutations that lead to upregulation of the *ampC* gene (Cabot et al, JAC 2018); upregulation of *ampC* was captured and may well cover variation in regulatory genes but it would be interesting to discuss why incorporation of SNPs in those genes did not strengthen the analysis. The Table itself would be improved by including a column on protein function, the Prokka/Roary names are not very informative.
8. Misclassification of isolates - inaccuracies in determining MICs (which is a well recognised problem, as discussed by the authors on p. 30) seems likely to be a contributing factor to this - were MICs measured more than once?
9. Page 26 line 12-14. The *prtN* gene is not (to the best of my knowledge) involved in pyocyanin production. Instead it is a regulator of production of pyocins, which is very different, and so these sentences should be altered. I also did not follow the possible connection between expression of *tra* genes and ciprofloxacin resistance - are the authors trying to explain why altered expression of *tra* genes is a predictor of resistance, did all *tra* genes have altered expression?
10. Page 28, lines 13-15. There is no obvious reason why upregulation of *gbuA* should relate to meropenem resistance - do the authors have any thoughts as to the connection? Similarly (line 23, same page) why would expression of *fpvA*, *pvdD* or *algF* relate to ceftazidime resistance/susceptibility? It would be good to discuss why genes/ proteins that have functions apparently unrelated to antibiotic resistance contribute to the model, especially when they are co-expressed with other genes (*alg* or *pvd*) for their normal function and the other genes apparently do not contribute to the model.
11. Materials and methods. The authors are to be commended for making their code and sequence data easily accessible to other researchers, though the DNA sequence reads are presumably embargoed until the research has been published. The section on SNP calling was not fully clear, does SAMtools call variants? Also, what are "heterozygous single nucleotide variants" in a haploid organism, does this mean where sequences in different reads did not concur?

1st Revision - authors' response

7 November 2019

Referee #1 (Comments on Novelty/Model System for Author):

I believe that the data presented is of sufficient quality, but requires proper controls (especially on expression data) to ensure that it is relevant for the task and for future reuse. Improving on that

would greatly improve the manuscript.

Referee #1 (Remarks for Author):

With antimicrobial resistance increasingly becoming a recognized public health threat, it is of great importance to improve the diagnostic tools available in the clinic, so that treatment decisions can be made quickly and precisely. This in turn would solve the problem of misuse of antibiotics, which is thought to be one of factors leading to the emergence of antimicrobial resistance. Using genetic and molecular markers has recently emerged as a promising way to accomplish this task, which could eventually become cost-effective as the price of recording molecular markers drops. In this manuscript, Khaledi, Weimann and colleagues use a collection of more than 400 *Pseudomonas aeruginosa* strains to test if a machine learning approach could be applied to predict resistance to four antimicrobials using both genetic and expression data. The authors report very high predictive power, with the most relevant features recapitulating known resistance mechanisms and claiming how expression data might be needed to obtain an effective predictor for three of the four antibiotics tested with their approach.

I appreciated reading this manuscript and I agree with the points raised in the introduction and discussion about the need for effective and quick diagnostic tests. I have however several concerns regarding the generation of the molecular data used in the predictors (specifically the expression data), the strategy used for training and main assessment of predictive power (specifically the influence of population structure) and with some of the terminology used, which is at times not following the standards found in the field of machine learning. These concerns are detailed below; while I think that some of those concerns are somewhat serious, I also believe that the authors can address them with minimal need for additional experimental work. I also want to congratulate the authors for choosing to make their software pipeline reproducible and transparent. I haven't had the time to properly inspect the code, but it looks appropriate at a superficial look.

Main concerns

* Expression data

While it seems like the expression data has been previously reported in a peer-reviewed publication (doi: 10.1038/s41396-019-0412-1), I was surprised to see how each strain was measured using a single technical and biological replicate. Could the authors comment on that and provide some indication on the reproducibility of the data, for instance by repeating the expression measurements for a few samples? The methods section also seems to not mention the potential for batch effects, which can arise if more than one sequencing run has been used or if different operators have performed RNA extraction at different times. Can the authors report more details of their experimental setup and what's the extent of potential batch effects on their data?

The reviewer is correct in that there are batch effects. They mainly depend on the sequencing runs and different experimentators responsible for harvesting the cells and extracting the RNA.

For two reasons, we are not concerned that this influences our results. First, our study focuses on a very large number of samples and we searched for significant accumulations of effects in phenotypically related groups of samples containing >100 isolates each (distributed across 24 independent sequencing pools). For most genes, batch effects are random and would therefore be leveled out in our analysis. Second, in the frame of another study we recorded the transcriptomes of 258 transposon mutants at least in duplicates for all samples. Those duplicates were always submitted to different sequencing runs. When analyzing these data we identified 137 particularly variable genes (none of which showed up in our biomarker list) and a fairly stable expression of the remaining part of the transcriptome. The figure below exemplary depicts the extent of correlation (pearson correlation coefficient) between 5 samples which can be seen as quintuplicates (3x PA14 wt (naturally LadS deficient) and 2x PA14-ladS tn mutant).

Additionally, a recent publication analyzing large scale *S. pyogenes* transcriptomic data (492 samples, 50 in triplicates) similarly concluded that there is very little variation between duplicates which would affect large scale analysis (Kacharoo et al, 2019, Nat Genet, doi: 10.1038/s41588-018-0343-1).

As suggested, we clarified the setup and relevance of batch-effects in the manuscript, page 25: “The 414 samples were distributed across 24 independent sequencing pools. We assessed possible batch effects using triplicates of the PA14-wt. The majority of the genome was very stably expressed across the replicates (pearson correlation coefficient ≥ 0.96).”

I am also curious to better understand which treatment of their input data has been fed to their machine learning models: from the methods section it looks like log-transformed counts were used, which might introduce a series of complications when used in machine learning models.

In fact, absolute transcription levels might not be indicative of the relevance of a gene towards resistance, and might introduce bias in the way the weights are assigned during model learning, especially when mixed together with the genetic data, which is binary in nature. Can the authors clarify exactly what transformation of the expression data was used in their model and whether using something like fold-changes with respect to the reference improves the model?

Reviewer 1 is right to assume that we use log-transformed counts; please see also Methods Section on RNA Sequencing : p 25 ll 16-17 “Expression counts were log-transformed (to deal with zero values we added one to the expression counts).” We agree that using the raw log-transformed expression values might be problematic, especially when mixing with the binary gene presence and absence or SNP features as machine learning models like the L1 regularized-SVM require features that have zero mean and variance in a similar magnitude. We therefore standardized the input features to zero mean and unit variance in any data type combination that included expression features. We explicitly mention now on pp 28-29 ll 25-3: “The expression features (EXPR) and any combination of features with another data type (GPA and SNPs) were transformed to have zero mean and unit variance, whereas binary features (GPA, SNPs and GPA+SNPs) were not transformed.” In fact, this should have a similar effect to using log fold changes with respect to the reference, which would center the expression values of each gene at the level of the respective PA14

gene.

* Model training and evaluation

Another major concern regards the way the machine learning models were trained and evaluated. Previous studies where machine learning approaches were used to predict antibiotics resistance have pointed out how population structure alone might sometimes be sufficient to deliver an accurate prediction (doi: 10.1101/403204), or how it "spills" in the genetic data (e.g. through gene presence/absence profiles, doi: 10.1371/journal.pcbi.1006258); furthermore, as samples are not independent from each other (especially if very closely related), great care has to be followed to define the held-out (or test) set. The main results showed by the authors (i.e. Figure 3) uses a test set that seems to have been generated by taking a random sample of all the strains used; while later the authors report having tested for the influence of population structure on their prediction (pages 21-22), they fail to report with sufficient emphasis the results (i.e. no figure). However, Figure 5 clearly shows how population structure might be influencing the models; in all instances, the "block cv" predictive performance is significantly lower than the "standard cv". The authors should better address this point and perhaps consider using a test set that is phylogenetically "insulated" from the training set (as done in *E. coli* doi: 10.1371/journal.pcbi.1006258).

We thank Reviewer 1 for this thoughtful comment. We followed this suggestion and devised a new test data set which was phylogenetically insulated from the training data: We only allowed strains with sequence types in this test data set that were not already part of the training data set (ll 10-12 p 14). Then, we re-trained the classifier on this data using as before a block cross-validation set-up. Reassuringly for the diagnostic classifiers we observed a comparable, but again more variable performance compared to the standard cross-validation estimates. We also note some exceptions to this observation for prediction of tobramycin resistance based fully or partly on SNPs (p 14 ll 10-22). We included these performance estimates in Figure 5. We now also emphasize the impact of population structure earlier (pp 7-8, ll 20-1) and also that the performance we obtained through block cross-validation is slightly worse for most data types (p 14, l 16), although not by much. In practice, sequence types will be much more mixed, so this can be seen as a worst case scenario. Sequence type information may even inform a better prediction based on the genetic background. The standard in the field is still to report only standard cross-validation results, which is why we picked a middle ground and reported results based on both types of cross-validation. We anticipate that in the future the field will move to solely reporting performance estimates based on block cross-validation.

While I greatly appreciated the data presented in Table 1, which clearly indicates how only a small subset of features is needed for an effective prediction, I would like to point out how the authors should better address the probable redundancy in their input data; in particular, gene presence/absence profiles and expression patterns are probably partly correlated, since an absent gene is likely to also not be expressed. Apart from that, previous works have tried to reduce the number of features when using bacterial genomics data (e.g. doi: 10.1371/journal.pcbi.1006258 and doi: 10.1186/1471-2164-13-170). Would the results change significantly if the input data was preconditioned to remove highly correlated features?

We thank Reviewer 1 for raising this excellent question. While we have removed completely redundant SNPs and gene presence/absence markers, we had not looked at correlation across data types. To tackle this, we calculated the point-bi-serial correlation coefficient to identify highly correlated expression and gene presence/absence features using a threshold of 0.9. We found that only 51 pairs of features were highly correlated, which corresponds to less than 1% of the coding genome of *Pseudomonas aeruginosa* PA14. This suggests that expression is apparently rarely completely uniform across all isolates.

Finally, I believe that the authors should correct the terminology used; specifically, they refer to a validation set (e.g. in Figure 5) when they probably meant test set (i.e. held-out set).

We agree that test data set is the more appropriate term in this context and have changed all such instances of this terminology in the text.

* Other concerns

The authors claim that expression data improves predictive power for three antibiotics out of four

tested; when looking at Figure 3 I however get the impression that expression data significantly improves prediction only for Ceftazidime. The authors should therefore correct their statement, unless some information is missing.

We appreciate the comment of Reviewer 2 and agree that this warranted further statistical scrutiny. Therefore, we went back to our results and used a one-sided t-test on the F1-macro scores from the five repeated cross-validation runs to investigate whether the classifiers including both gene presence and absence, and expression were actually better than just using the expression profiles. For all three drugs meropenem, ceftazidime and tobramycin we found that the expression data significantly improved the F1-macro score. See p 10 ll 14-15, 25 and p 11 ll 6.

Minor comments

* Page 6: "common anti-pseudomonas antimicrobials, tobramycin (TOB), ceftazidime (CAZ), ciprofloxacin (CIP), and meropenem (MEM)", can the authors provide a reference for this statement?

References were added on page 6 ll 21-22.

* Figure 1: it's difficult to read the colors on the phylogenetic tree, even on a screen

We have addressed this as suggested and provided a color-revised Figure 1.

* Figure 3: using many different metrics to measure predictive power could perhaps be relegated to a supplementary figure (the metrics seem to be highly correlated anyway); changing this figure to be have a similar layout as Figure 5 (using only the F1-score) might help readability

We agree with Reviewer 2. We now have removed the measures for the predictive values from the Figure as they correlate quite strongly with the sensitivity of the resistance and susceptibility classes. We think this improves the visibility. We continue to refer to the numbers for the predictive values of both classes in the text.

* I couldn't find the genomic data on either SRA/ENA, even though the authors provided an accession number (PRJNA526797). Is it currently under an embargo?

The DNA sequencing data will be publically available, as soon as the manuscript is published. There is already limited access for now:
<https://dataview.ncbi.nlm.nih.gov/object/PRJNA526797?reviewer=r0nrtgs3k8mdmjv0gpik0h57fn>

* I couldn't find the input data used for training and prediction in the provided repositories: could the authors provide them in the next release, so that different approaches could be tested?

We support data sharing with the scientific community to enable further analysis based on the genomic and phenotypic data including any intermediate results that we have generated. In addition to the GitHub repository accompanying this paper we have uploaded the feature tables and resistance data used as input to the machine learning to a Zenodo repository (Section Data Availability p 30 ll 23-25), DOI: 10.5281/zenodo.3464542).

Referee #2 (Remarks for Author):

The authors have carried out a very comprehensive study, incorporating genomic and transcriptional data on a very large number of isolates of *P. aeruginosa* and using machine learning, in order to improve tools for prediction of antibiotic resistance in this species. The approaches used by the authors, in particular the integration of gene expression analysis with genome sequence analysis but also the use of machine learning in the context of *P. aeruginosa* antibiotic resistance, are novel and give important new insights. In addition the authors have been very rigorous in developing and validating the approaches that they used. Consequently they have developed a tool that can predict resistance with high accuracy. My only real reservation regarding the manuscript is that a number of the genes revealed by machine learning as being good predictors of resistance/sensitivity are non-

intuitive (and one or two that might be expected do not appear). This may well reflect a lack of knowledge of the basis of antibiotic resistance in *P. aeruginosa* rather than a flaw in the study design, but the manuscript would be strengthened through more discussion of the genes uncovered in the analysis. The article is generally clearly written but some parts especially the machine learning sections (p.8-14) would be challenging for much of the target audience (molecular biologists, clinicians and medical doctors).

We appreciate the concerns Reviewer 2 raises with respect to the readability of the machine learning section. While we agree that this section may seem quite technical to a non-specialist audience, we feel it is nevertheless important to include the details of the validation in the results. We have tried to improve the text to make it slightly more understandable for a more general audience.

Additionally, we have adapted this section in response to the specific concerns voiced above and below, in terms of terminology (test versus validation set), block cross-validation and relation between the C hyperparameter and model sparsity (Figure 4). Furthermore we discuss the identified biomarkers now in more detail.

1. Title. In my opinion the title does not accurately reflect the content of the manuscript, they are not "fighting resistance". The authors have used machine learning to develop tools for understanding the basis of antibiotic resistance in *Pseudomonas aeruginosa*, extending to predicting resistance, and in my opinion a title along those lines would better reflect the manuscript contents. Using the tools developed here to fight resistance (or at least to provide better targeted treatment) is the next step!

The title was adapted accordingly (Predicting resistance...).

2. Figure 1 shows that the authors have a very extensive diversity of isolates which is essential for the study. Some of the isolates were phylogenetically very close which runs the risk of mis-identifying resistance- or sensitivity- associated SNPs due to co-inheritance in clonal isolates. The authors' "block" approach showed that this was not a problem, but it was a bit confusing that this is shown diagrammatically in Fig 2 but not explained until p. 22.

We thank Reviewer 2 for this suggestion. We have now added a sentence in the Results on to introduce the concept of block cross-validation earlier in the text.: "... we also assessed performance while accounting for population structure based on sequence types through a block cross-validation approach (pp 7-8 ll 21-1)."

3. Along similar lines - Fig 1C shows the frequencies of resistance of isolates to different antibiotics. For isolates with multiple resistances was there a bias in co-resistances? Eg. for bacteria that were meropenem resistant, was there a disproportionate number that were also ceftazidime-resistant? A bias in co-resistance could mislead association between genome markers, or expression, and resistance to a given antibiotic.

We thank the reviewer for the great comment. We investigated shared information across drugs and describe the results in the supplement (Figure S7), in order to better address the comment and emphasized our findings in the discussion section (p 33). Despite indeed widely overlapping resistances, we found only very little overlap in the identified markers. E.g. for ceftazidime, tobramycin and meropenem the best performing diagnostic classifiers (= those using both expression and gene presence/absence features), we found only overlapping markers between ceftazidime and tobramycin at all. These were expression of PA14_15420 and presence of A7J11_02078/sul1/foIP_2, group_282, group_3462 and group_5517 which account for 5/59 and 5/37 of the total features or 14.7%/17.1% of the total weight of the ceftazidime and tobramycin SVM classifiers, respectively. Group_282, group_3462 and group_5517 genes are hypothetical genes. sul1 which is located on mobile elements (usually class I integrons) could indicate that the shared signal of the tobramycin and ceftazidime classifiers is due to resistance genes for both drugs, being found on the same resistance cassettes, as class I integrons carrying betalactamases as well as aminoglycoside modifying enzymes are frequently detected (examples: Poirel et al., 2001, AAC; Fonseca et al., 2005, FEMS Immunol Med Microbiol) (see p 21 ll 17-25 and p 20 ll 1-5).

4. Page 8 line 13-14. Having a portion of the isolates as a training set and a portion as the validation set is very logical, but how was the ratio of 80:20 decided on, did the authors investigate other

ratios?

The number chosen was a compromise between having a large training set (as large as possible, to obtain the best model), and a separate test set of sufficient size to be informative. It is common in the field to use 10-20% of the data as held-out data set. Generally with increasing sample set sizes, performance estimates become less variable, and thus more informative, which is important for experiments without replicates, such as this one.

5. Figure 3 and Figure 5 - the multiple box plots and colours make this difficult to see in detail. The authors could try having black lines with coloured fill for the boxes.

We agree that Figure 3 and especially Figure 5 are a bit cluttered. For Figure 5 we now omitted the predictive value of the resistance and susceptibility class, as those measures are highly correlated with the sensitivity of the resistance and susceptibility class. We hope that this improves clarity. We still mention individual numbers of the predictive values measures in the text. We tried using black lines with coloured fill as suggested by Reviewer 2, but this did not improve clarity in our view.

6. Figure 4. This is one example where the non-specialist may struggle (this one did). It would be helpful to expand the explanation of the C parameter and the role of regularization. How do the data show that the ciprofloxacin classifier needed only 2 SNPs to saturate the learning curve whereas the other antibiotics needed 50 or more? - I can make a guess but it would be helpful to explain.

We apologize for not being clear enough in the first version of the manuscript. Predicting antimicrobial resistance from genomic features comes with a very large number of genomic features and comparably few data points, here the isolates, which is known as a “low n, high p problem”, or as short, fat data. Regularization provides a crucial way to explore predictors that include a varying number of discriminatory features. Specifically for the SVM, the C parameter penalizes the total contribution of features, i.e. the sum of the features weights in the optimization problem. By measuring performance of more or less sparse models via cross-validation, we can then pick the predictor providing the best performance, with the lowest number of features. In Figure 4 we can see that the C parameter is inversely related to the number of markers being included in the model i.e. lower values for the C parameter yield models with less features. For each value of the C parameter, we recorded the performance of the predictor (Panel B) and how many features were included in the models (Panel A). For ciprofloxacin the Figure shows that very few features are sufficient to obtain a near optimal performance. The performance curve is flat and only begins to increase slightly when we explore much higher values for the C parameter. For the other diagnostic classifier we see a more or less steady increase before we reach the optimal performance. For more clarity, we have also added a sentence to the legend of Figure 4 (p 39, ll 23-25).

7. Table 2 and associated analyses. This contains a few surprises. For example, with tobramycin, a recent paper involving one of the authors indicated that *fusA1* was a major determinant of tobramycin resistance in *P. aeruginosa* (Lopez-Causape et al, AAC, 2018). It therefore seems surprising that *fusA1* SNPs did not strengthen the analysis. Similarly, ceftazidime resistance is associated with mutations that lead to upregulation of the *ampC* gene (Cabot et al, JAC 2018); upregulation of *ampC* was captured and may well cover variation in regulatory genes but it would be interesting to discuss why incorporation of SNPs in those genes did not strengthen the analysis. The Table itself would be improved by including a column on protein function, the Prokka/Roary names are not very informative.

The Roary output provided some further information on the gene name and protein function, which we have added to Table 2 (notably *oprD_1*, *pknk_1*), and Supplementary Table 5 (columns Roary non-unique name, Roary annotation; pp 42-43 ll 27-2).

We had a closer look into why we did not capture previously described mutations in regulatory proteins (that e.g. impact on *ampC* expression). Constitutive *ampC* up-regulation can be triggered by a great variety of single mutational adaptations (e.g. in *AmpD* and *AmpD* homologues, *AmpG*, *AmpR*, the *ampR-ampC* intergenic region or penicillin binding proteins such as *PBP4*).

Indeed (as outlined in the revised discussion, p 20 ll 12-15, p 21, ll 1-13) we identified a small number of isolates in the resistant group (11 of 165) harboring a R504C substitution in the gene

ftsI†(PBP3), known to confer to ceftazidime resistance. However, these isolates all exhibited a strong ampC†overexpression, most likely explaining why ftsI†was not identified as an additional discriminative marker in our analysis.

We also had a closer look into the fusA1 I61M substitution (main mutation found by Lopez-Causape et al, AAC, 2018). None of our clinical isolates harbored this mutation. This might be due to the fact, that the observation by Lopez-Causape et al was based on in vitro evolution. We did find various other fusA1 mutations. However, they were distributed over most of the gene without any particular accumulation at a specific site in the resistant group of isolates (4 SNPs at the same position at the most). This is most likely the reason why they were not identified as strong discriminators in our machine learning analysis.

8. Misclassification of isolates - inaccuracies in determining MICs (which is a well recognised problem, as discussed by the authors on p. 30) seems likely to be a contributing factor to this - were MICs measured more than once?

All MICs were tested in triplicates. In the case of varying results up to five measures per isolate were performed. Isolates with not at least three matching results were excluded from our study. This is now clarified in the methods section (p 24, ll 14-16).

9. Page 26 line 12-14. The prtN gene is not (to the best of my knowledge) involved in pyocyanin production. Instead it is a regulator of production of pyocins, which is very different, and so these sentences should be altered. I also did not follow the possible connection between expression of tra genes and ciprofloxacin resistance - are the authors trying to explain why altered expression of tra genes is a predictor of resistance, did all tra genes have altered expression?

We apologize for this mistake and of course corrected the connection of prtN to pyocins. Furthermore, the information on the tra genes was removed as this was indeed very speculative (p 17, ll 12-22).

10. Page 28, lines 13-15. There is no obvious reason why upregulation of gbuA should relate to meropenem resistance - do the authors have any thoughts as to the connection? Similarly (line 23, same page) why would expression of fpvA, pvdD or algF relate to ceftazidime resistance/ susceptibility? It would be good to discuss why genes/ proteins that have functions apparently unrelated to antibiotic resistance contribute to the model, especially when they are co-expressed with other genes (alg or pvd) for their normal function and the other genes apparently do not contributors to the model.

We extended the discussion on gbuA and the seemingly unrelated gene expression markers fpvA, pvdD and algF in the manuscript (p 19, ll 21-24 and p 20, ll 12-14).

11. Materials and methods. The authors are to be commended for making their code and sequence data easily accessible to other researchers, though the DNA sequence reads are presumably embargoed until the research has been published. The section on SNP calling was not fully clear, does SAMtools call variants? Also, what are "heterozygous single nucleotide variants" in a haploid organism, does this mean where sequences in different reads did not concur?

Yes, the reviewer is correct. We clarified the expression "heterozygous single nucleotide variants" in the revised version of the manuscript. SAMtools was used for variant calling from the Stampy pile up. However, we noticed that calling errors occurred at positions with varying nucleotides (e.g. due to sequencing errors). As an example, we found cases where a SNP was called, although only less than 10 % of the reads contained the respective nucleotide change to the reference. Likewise, we found that some "true" SNPs (e.g. in >90 % of the reads) located close to insertion or deletion sites were not called. Thus, we implemented an additional step for correction of these obvious errors where the respective positions were converted into the most likely sequence according to the most frequently occurring nucleotide at this position. This is now clarified in the manuscript (p 26, ll 14-20).

2nd Editorial Decision

10 December 2019

Thank you for the submission of your revised manuscript to EMBO Molecular Medicine. We have now received the enclosed reports from the referees that were asked to re-assess it. As you will see, the reviewers are now globally supportive and I am pleased to inform you that we will be able to accept your manuscript pending the following final amendments:

Please address the comments of referee 1. You should clearly state that blocking CV leads to a significant drop in predictive power, while it is now written as a negligible difference, which is not the case according to this referee. Please amend.

I look forward to reading a new revised version of your manuscript as soon as possible.

***** Reviewer's comments *****

Referee #1 (Remarks for Author):

I appreciate the work put in by the authors in addressing the reviewers' questions. I believe that my concerns were mostly satisfied, but I would like to point out a couple of things that must be addressed further to make this manuscript worthy of publication.

RNA-seq: I appreciate the authors showing some of their own data to support the absence of replicates, as well as citing interesting recent literature. I would suggest adding an additional supplementary figure with the triplicate reference strain to help convince the reader as well.

Block CV: The authors added an analysis on using a "phylogenetically insulated" test set, and wrote on page 14: "Overall, the performance estimates we obtained using the phylogenetically insulated test dataset were comparable to the cross-validation estimates, only tobramycin resistance prediction using classifiers trained fully or partly on SNPs dropped in performance.". When looking at Figure 5 and comparing block CV vs standard CV it is very clear that there is a difference and that it is most certainly more significant than the one shown in Figure 3, for which the authors are now reporting a t-test p-value. I think the authors should comment on this relatively large drop in predictive power, which is probably due to either overfitting or genetic background effects, or a combination of both. The authors do recognize in their rebuttal that future studies will eventually use only block CV when assessing predictions, so it only make sense that they mention in the manuscript why that is indeed important.

I thank the authors for providing the input data as a Zenodo repository. I had some difficulty loading it, so I would suggest adding a brief README with some minimal instructions on how to read the sparse matrices in numpy. The data looks otherwise very well formatted and definitely very useful for the community.

Referee #2 (Comments on Novelty/Model System for Author):

Reasons are the same as for my initial review

Referee #2 (Remarks for Author):

The authors have addressed all the points from my earlier review in a satisfactory manner. Thank you for the clarifications.

2nd Revision - authors' response

24 December 2019

Referee #1 (Remarks for Author):

I appreciate the work put in by the authors in addressing the reviewers' questions. I believe that my concerns were mostly satisfied, but I would like to point out a couple of things that must be addressed further to make this manuscript worthy of publication.

RNA-seq: I appreciate the authors showing some of their own data to support the absence of replicates, as well as citing interesting recent literature. I would suggest adding an additional supplementary figure with the triplicate reference strain to help convince the reader as well.

We added an additional supplementary figure (Appendix Figure S3) including the correlation of the triplicate reference strain.

Block CV: The authors added an analysis on using a "phylogenetically insulated" test set, and wrote on page 14: "Overall, the performance estimates we obtained using the phylogenetically insulated test dataset were comparable to the cross-validation estimates, only tobramycin resistance prediction using classifiers trained fully or partly on SNPs dropped in performance.". When looking at Figure 5 and comparing block CV vs standard CV it is very clear that there is a difference and that it is most certainly more significant than the one shown in Figure 3, for which the authors are now reporting a t-test p-value. I think the authors should comment on this relatively large drop in predictive power, which is probably due to either overfitting or genetic background effects, or a combination of both. The authors do recognize in their rebuttal that future studies will eventually use only block CV when assessing predictions, so it only make sense that they mention in the manuscript why that is indeed important.

We thank the reviewer for his/her remark. The fact that we did not clearly distinguish between the description of the results based on the phylogenetically insulated test dataset and the results based on the block cross-validation may have added some confusion. We meant to say that the block crossvalidation results showed decreased performance compared to the standard cross-validation "Overall, the performance estimates we obtained using the phylogenetically insulated test dataset were comparable to the [block] cross-validation estimates, only tobramycin resistance prediction using classifiers trained fully or partly on SNPs dropped in performance." In contrast, the performance estimates we obtained via the block cross-validation and through the corresponding phylogenetically insulated dataset were indeed similar except for data type combination that we explicitly mention. We have now restructured the description of these particular results.

"In addition, instead of using a random assignment of strains into test and training dataset, we analyzed the performance only allowing strains in a test dataset corresponding to the block crossvalidation training dataset with sequence types that were not already included in this training dataset. For all classifiers including our candidate diagnostic classifiers, we found that the block cross-validation performance estimates were slightly lower than those obtained using a sequence type unaware estimation (F1-score difference between ~0.03 and 0.05 for the diagnostic classifiers). This was particularly apparent for some suboptimal data type combinations, such as for predicting tobramycin resistance using SNPs or gene expression, where a substantially lower discriminative performance was achieved in block- compared to random cross-validation (macro F1-score difference > 0.1, Dataset EV3). Interestingly, we observed that the ranking of the performance by data type remained almost identical for all drugs. Overall, the performance estimates we obtained using this phylogenetically insulated test dataset were comparable to the block cross-validation estimates, only tobramycin resistance prediction using classifiers trained fully or partly on SNPs dropped considerably in performance. "

I thank the authors for providing the input data as a Zenodo repository. I had some difficulty loading it, so I would suggest adding a brief README with some minimal instructions on how to read the sparse matrices in numpy. The data looks otherwise very well formatted and definitely very useful for the community.

We are glad that Reviewer 1 appreciated our efforts to provide the processed sequencing and phenotyping data in a public data repository. We now have also added a brief README (README.md) in the second version of the same data repository including some instruction on how to read the Numpy sparse matrices.

Corresponding Author Name: Susanne Häussler, Alice C. McHardy

Manuscript Number: EMM-2019-10264